# NIERT: Accurate Numerical Interpolation through Unifying Scattered Data Representations using Transformer Encoder

## Abstract

Numerical interpolation for scattered data aims to estimate values for target points based on those of some observed points. Traditional approaches produce estimations through constructing an interpolation function that combines multiple basis functions. These approaches require the basis functions to be pre-defined explicitly, thus greatly limiting their applications in practical scenarios. Recent advances exhibit an alternative strategy that learns interpolation functions directly from observed points using machine learning techniques, say deep neural networks. This strategy, although promising, cannot effectively exploit the correlations between observed points and target points as it treats these types of points separately. Here, we present a learning-based approach to numerical interpolation using encoder representations of Transformers (thus called NIERT). NIERT treats the value of each target point as a masked token, which enables processing target points and observed points in a unified fashion. By calculating the partial self-attention between target points and observed points at each layer, NIERT gains advantages of exploiting the correlations among these points and, more importantly, avoiding the unexpected interference of target points on observed points. NIERT also uses the pre-training technique to further improve its accuracy. On three representative datasets, including two synthetic datasets and a real-world dataset, NIERT outperforms the existing approaches, e.g., on the TFRD-ADlet dataset for temperature field reconstruction, NIERT achieves an MAE of $1.897 \times 10^{-3}$, substantially better than the transformer-based approach (MAE: $27.074 \times 10^{-3}$). These results clearly demonstrate the accuracy of NIERT and its potential to apply in multiple practical fields.

## 1 Introduction

Numerical interpolation for scattered data plays important and fundamental roles in a wide range of practical scenarios, including solving partial differential equations (PDEs) [1], temperature field reconstruction [2], time series interpolation [3, 4]. In meshfree PDE solvers, the interpolation error often leads to deviations in subsequent calculations, which seriously affects the solution's accuracy [5]. In the task of temperature field reconstruction for micro-scale electronics, interpolation methods are used to obtain the real-time working environment of electronic components from limited measurements, and imprecise interpolation will significantly increase the cost of predictive maintenance [2]. Thus, accurate approaches to numerical interpolation are highly desirable.

A large number of approaches have been proposed for interpolation of scattered data, which can be divided into two categories, namely, traditional non-learning based methods and recent learning-based methods. The typical traditional interpolation schemes construct the target function by a linear

combination of basis functions [6]. These schemes require explicitly-defined basis functions to model the target function space, and various types of basis functions have been devised by algorithm designers to adapt to different scenarios. Nevertheless, such methods still suffer the from limitations of high requirement of sufficient observed points, and the limited complexity of the target function.

Recent progress has exhibited an alternative strategy that uses neural networks to learn interpolation functions directly from the given observed points. For example, conditional neural processes (CNPs) [7] and their extensions [8–10] model the conditional distribution of regression functions given the observed points. In addition, Chen et al. [2] proposed to use vanilla Transformer [11] to solve interpolation task in temperature field reconstruction. All of these approaches use an "encoder-decoder" architecture, in which the encoder learns the representations of observed points while the decoder estimates values for target points. Intuitively, observed points and target points should be processed in a unified fashion because they are from the same domain. However, this architecture treats them separately and cannot effectively exploit the correlation between them.

Inspired by the recent advances of language/image models, especially BERT [12] and BEiT [13], we designed an approach to numerical interpolation that can effectively exploit the correlations between observed points and target points. Our approach is a learning-based approach using the encoder representations of Transformers (thus called NIERT). The key elements of NIERT include: $i$) the use of the mask mechanism, which enables processing target points and observed points in a unified fashion, $ii$) a novel partial self-attention model, which calculates attentions between target points and observed points at each layer, thus gaining the advantages of exploiting the correlations between these two types of points and, more importantly, avoiding the unexpected interference of target points on observed points simultaneously, and $iii$) the use of the pre-training technique, which further improves the interpolation accuracy of NIERT.

The main contributions of this study are summarized as follows.

1. We propose an accurate approach to numerical interpolation for scattered data. On representative datasets, including both synthetic and real-world datasets, our approach outperforms existing approaches. The experimental results demonstrate the potential of our approach in a wide range of application fields. The source code of NIERT will be released for open source use.

2. We propose a novel partial self-attention mechanism to make Transformer incorporated with strong inductive bias for interpolation tasks; i.e., it can effectively exploit the correlation among two types of points but simultaneously avoid the interference of one type of points onto the others.

3. We propose to use the pre-training technique to enhance interpolation approaches. When facing an interpolation task in a newly-appearing application field, we can benefit from the experience learned from low-cost synthesized interpolation tasks.

## 2 Related works

### 2.1 Traditional interpolation approaches for scattered data

Traditional interpolation approaches for scattered data use explicit basis functions to construct interpolation function, e.g., Lagrange interpolation, Newton interpolation [6], B-spline interpolation [14], Shepard's method [15], Kriging [16], and radial basis function interpolation (RBF) [17, 18]. Among these approaches, the classical Lagrange interpolation, Newton interpolation and B-splines interpolation are usually used for univariate interpolation. Wang et al. [19] proposed a high order multivariate approximation scheme for scattered data sets, in which approximation error is represented with Taylor expansions at data points, and basis functions are determined through minimizing the squares of approximation error.

### 2.2 Neural network-based interpolation approaches

Equipped with deep neural networks, data-driven interpolation and reconstruction methods show great advantages and potential. For instance, convolutional neural networks (CNNs) have been applied in the interpolation tasks of single image super-resolution [20, 21], and recurrent neural networks (RNNs) and Transformers have been used for interpolation of sequences like time series data [4, 22].

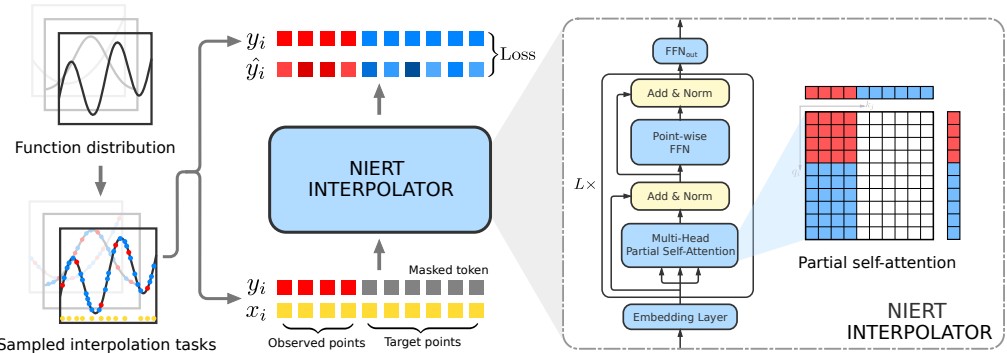

Figure 1: Overview of NIERT training process. Here, $x_i$ represents the position of a point, and $y_i$ represents its value. The predicted values of the point is denoted as $\hat{y}_i$. We prepare the training interpolation tasks by first sampling functions from a function distribution $\mathcal{F}$ and then sampling observed points $O$ and target points $T$ on each function. NIERT trains an interpolator over this data. The partial self-attention mechanism facilitates exploiting the correlations between observed points and target points and avoiding the unexpected interference of target points on observed points

Recently, Garnelo et al. [7] proposed to model the conditional distribution of regression functions given observed points. The proposed approach, conditional neural processes (CNPs), has shown increased estimation accuracy and generalizing ability. Kim et al. [8] designed an enhanced model, attentive neural processes (ANPs), with improved accuracy. Lee et al. [10] leveraged Bayesian last layer (BLL) [23] for faster training and better prediction. In addition, the bootstrap technique was also employed for further improvement [9]. To solve the interpolation task in 2D temperature field reconstruction, Chen et al. [2] proposed an Transformer-based approach, referred to as TFR-tranformer, which can also be applied to solve interpolation tasks for scattered data with higher dimensions. Note that although TFR-transformer and our NIERT are both based on transformer, they are fundamentally different: $i$) TFR-transformer adopts an encoder-decoder structure and treats observed points and target points respectively, while NIERT adopts only a Transformer encoder (equiped with partial self-attention) to encode and learn corelatetion of the scattered data in a unified fashion; $ii$) TFR-transformer is trained by minimizing the prediction error of target points, while NIERT's training objective considers the prediction error of both observation points and target points.

## 2.3 Masked language/image models and the pre-training technique

The design of NIERT is also inspired by the recent advances in masked language/image models [12, 13, 24, 25] and pre-trained models for symbolic regression [26, 27]. The masked pre-trained models have been shown to be successful in learning representations of languages and images from large-scale data and improving the performance of downstream tasks. In addition, the mask mechanism makes the model able to reconstruct missing data from their context. Utilizing large-scale synthetic symbolic functions and sampled scattered data, Biggio et al. [26] and Valipour et al. [27] pre-trained Transformers to learn the map from scattered data to corresponding symbolic formulas. Different from these approaches, NIERT uses synthetic data to learn interpolating functions from scattered data numerically.

# 3 Method

## 3.1 Overview of NIERT

In the study, we focus on the interpolation task that can be formally described as follows: We are given $n$ observed points with known values $O = \{(x_i, y_i)\}_{i=1}^n$, and $m$ target points with values to be determined, denoted as $T = \{x_i\}_{i=n+1}^{n+m}$. Here, $x_i \in X$ denotes position of a point, $y_i = f(x_i) \in Y$ denotes the value of a point, and $f : X \to Y$ denotes a function mapping positions to values. The function $f$ is from a function distribution $\mathcal{F}$, which can be explicitly defined using a mathematical formula or implicitly represented using a set of scattered data in the form $(x_i, y_i)$. The goal of

**Algorithm 1** NIERT training process

---

**Require:** NIERT model $M_\theta$ with parameters $\theta$, epoch number $N$, batch size $B$, domain $X$, function distribution $\mathcal{F}$ and error metric function $\mathrm{Error}(\cdot, \cdot)$

**for** $k$ in $\{1..N\}$ **do**
    $J \leftarrow 0$
    **for** $b$ in $\{1..B\}$ **do**
        $f, \{x_i\}_i \leftarrow$ sample a function and scatter points from $\mathcal{F}, X$
        $\{y_i\}_i \leftarrow$ calculate $f$ on $\{x_i\}_i$
        $O, T \leftarrow$ split and mask scatter points with values $\{(x_i, y_i)\}_i$
        $\{\hat{y}_i\}_i \leftarrow M_\theta(O, T)$
        $J \leftarrow J + \sum_i \mathrm{Error}(\hat{y}_i, y_i)$
    **end for**
    Compute the gradient $\nabla_\theta J$ and update $\theta$
**end for**

---

interpolation task is to accurately estimate the values $f(x)$ for each target point $x \in T$ according to the observed points in $O$.

Figure 1 depicts the schematic diagram of our NIERT approach. Briefly speaking, our approach employs a data-driven approach to numerical interpolation using encoder representations of Transformers. The main element of our approach is a neural interpolator that learns to estimate values for target points. The interpolator is featured by the characteristic that it treats the value of each target point as a masked token, thus enabling the unifying fashion to process both target points and observed points in the subsequent encoding and estimation procedures.

To suit the interpolation task, we design a *partial self-attention* mechanism: on one side, we calculate the attention between target points and observed points at each layer, which gains NIERT the advantage to effectively exploit the correlations between these two types of points. On the other side, the attention is a partial one as we do not consider the effects of a target point on all other points. This way, the unexpected interference of target points onto the observed points, and the interference among target points, are completely avoided.

The training process is depicted in Algorithm 1. Specifically, we prepare the training interpolation tasks by first sampling functions from a distribution $\mathcal{F}$ and then sampling observed points $O$ and target points $T$ on each function. When training NIERT, we set the loss function as the error between the estimated values and the corresponding ground-truth. It should be pointed out that errors acquired on both observed points and target points are accounted into loss function.

### 3.2 Architecture of the NIERT interpolator

The neural interpolator in NIERT adopts the Transformer encoder framework; however, to suit the interpolation task, significant modifications and extensions were made in embedding, Transformer and output layers, which are described in details below.

**Embedding with masked tokens:** NIERT embeds both observed points and target points into the unified high-dimensional embedding space. As the position $x$ of a data point and its value $y$ are from different domains, we use two linear modules : $\mathrm{Linear}_x$ embeds the positions while $\mathrm{Linear}_y$ embeds the values.

It should be noted that for target points, their values are absent when embedding as they are to be determined. In this case, we use a masked token as substitutes, which is embedded as a trainable parameter $\mathrm{MASK}_y$ as performed in BERT [12]. This way, the interpolator processes both target points and observed points in a unifying fashion.

We concatenate the embeddings of position and value of a data point as the point's embedding, denoted as $h_i^0$, i.e.,

$$h_i^0 = \begin{cases} [\mathrm{Linear}_x(x_i), \mathrm{Linear}_y(y_i)], & \text{if } (x_i, y_i) \in O \\ [\mathrm{Linear}_x(x_i), \mathrm{MASK}_y], & \text{if } x_i \in T \end{cases}$$

**Transformer layer with partial self-attention mechanism:** NIERT feeds the embeddings of the points into a stack of $L$ Transformer layers, producing encodings of these points as results. Each Transformer layer contains two subsequent sub-layers, namely, a multi-head self-attention module, and a point-wise fully-connected network. These sub-layers are interlaced with residual connections and layer normalization between them.

To avoid the unexpected interference of target points on observed points and target points themselves, NIERT replaces the original self-attention in Transformer layer with a *partial self-attention*, which calculates the feature of point $i$ at the $l+1$-st layer as follows:

$$h^{l+1} = \text{LayerNorm}(\widetilde{v}^l + \text{MLP}(\widetilde{v}^l)),$$

$$\widetilde{v}_i^l = \text{LayerNorm}\Big(v_i^l + \sum_j w_{ij}\alpha_{ij}^l v_j^l\Big)$$

where $v_i^l$ and $\alpha_{ij}^l$ represent the value embedding and ordinary attention weights at the $l$-th layer as calculated in Transformer [11]. In this formula, we introduce a new term $w_{ij}$ that represents the partial self-attention pattern, i.e.,

$$w_{ij} = \begin{cases} 1, & \text{if } (x_j, y_j) \in O \\ 0, & \text{if } x_j \in T \end{cases}.$$

By forcing the weight $w_{ij}$ to be 0 for a target point $i$ and any point $j$, we completely avoid the unexpected interference of target points on the other points.

**Estimating values for target points:** For each target point $i$, we estimate its value $\hat{y}_i$ through feeding its features at the final Transformer layer into a fully connected feed-forward network, i.e.,

$$\hat{y}_i = \text{MLP}_{\text{out}}(h_i^L).$$

We calculate the error between the estimation and the corresponding ground-truth value, and compose the errors for all points into a loss function to be minimized.

### 3.3 Enhancing NIERT using pre-training technique

The interpolation functions from different applications usually differ greatly in their forms; however, the interpolation tasks might still share some common characteristics, say the correlation between observed points and target points. These common characteristics enable enhancing NIERT using the pre-training technique. Here, we pre-train NIERT using a synthetic dataset (see 4.1 for further details) and fine-tune it on other datasets in application fields.

## 4 Experiments and results

We evaluated NIERT and compared it with ten representative scattered data interpolation approaches on both synthetic and real-world datasets. We also examined the effects of the key elements of NIERT, including the partial self-attention, and the pre-training technique.

### 4.1 Experiment setting

The datasets, metrics and approaches for comparison are briefly described below. Further details of experiment settings are provided in Supplementary text.

**Datasets:** Three representative datasets in various application fields, including two synthetic datasets NeSymReS and TFRD-ADlet, and real-world dataset PhysioNet, are used for evaluation.

NeSymReS is a synthetic dataset for mathematical function interpolation, which is built using data generator proposed by Biggio et al. [26] and Lample and Charton [28]. We construct a function set with various dimensionality of data points, including 1D, 2D, 3D, and 4D. Scattered points in each instance are randomly sampled and divided into observed points and target points.

TFRD-ADlet [2] is a synthetic dataset for 2D temperature field reconstruction where each instance represents a simulated 2D temperature field containing several heat source components and a specific

Dirichlet conditioned boundary. The goal of each task instance is to reconstruct the whole temperature field according to a limited number of observed points with measured temperature.

PhysioNet Challenge 2012 dataset [29] is a real world dataset collected from intensive care unit (ICU) records for time-series data interpolation. Each point in an instance represents a measurement at a specific time, where each measurement contains up to 37 physiological indices. Following the study [22], we randomly divided the points into observed points and target points by setting the ratio of observed points at five levels, i.e., 50%, 60%, 70%, 80%, and 90%. Note that this dataset is a representative of hard interpolation tasks due to the sparsity and irregularity of the records.

**Pre-training dataset:** In this study, NeSymReS dataset was used for pre-training NIERT to further improve its interpolation accuracy on TFRD-ADlet and PhysioNet dataset. For TFRD-ADlet dataset we directly use 2D TFRD-ADlet dataset for pre-training. As the PhysioNet dataset has a dimensionality of 37, we construct the pre-training instances by stacking random 37 functions from 1D TFRD-ADlet dataset and then sampling interpolation task instances.

**Metrics:** When evaluating NIERT and other interpolation approaches, the prediction error of target points are calculated as interpolation accuracy. For the NeSymReS and PhysioNet dataset, we adopted mean squared error (MSE) as the error metric. For TFRD-ADlet dataset, we use three error metrics: mean absolute error (MAE), MAE in component area (CMAE) and MAE at boundary (BMAE) following Chen et al. [2]. Accordingly, we use $L_2$-form loss function for NeSymReS and PhysioNet dataset and $L_1$-form loss function for TFRD-ADlet dataset for training.

**Approaches for comparison:** For NeSymReS dataset, we compared NIERT with six representative interpolation approaches, including RBF[30], MIR [19], CNP[7], ANP[8], BANP[9] and TFR-transformer [2]. For TFRD-ADlet we compared NIERT with CNP, ANP, BANP and TFR-transformer. For PhysioNet we compared NIERT with four approaches designed for time-series data interpolation, including RNN-VAE [31], L-ODE-RNN [32], L-ODE-ODE[33], and mTAND-Full[22].

## 4.2 Interpolation accuracy on synthetic and real-world datasets

For each instance of the test datasets, we applied the trained NIERT to estimate values for the target points. We calculate the errors between the estimation and the ground-truth as interpolation accuracy.

| Interpolation approach | MSE ($\times 10^{-5}$) on NeSymReS test set | | | |
|---|---|---|---|---|
| | 1D | 2D | 3D | 4D |
| RBF | 215.439 | 347.060 | 443.094 | 327.775 |
| MIR | 67.281 | 274.601 | 448.933 | 342.997 |
| CNP | 67.176 | 248.668 | 392.348 | 314.311 |
| ANP | 34.558 | 140.005 | 206.699 | 164.751 |
| BANP | 14.913 | 84.187 | 143.518 | 140.288 |
| TFR-transformer | 15.556 | 58.569 | 99.986 | 90.579 |
| NIERT | **8.964** | **45.319** | **77.664** | **72.025** |

Table 1: Interpolation accuracy of NIERT and the existing approaches on NeSymReS test dataset

| Interpolation approach | Evaluation criteria ($\times 10^{-3}$) | | |
|---|---|---|---|
| | MAE | CMAE | BMAE |
| CNP | 96.674 | 109.419 | 56.939 |
| ANP | 54.684 | 62.511 | 26.524 |
| BANP | 28.671 | 29.450 | 19.984 |
| TFR-transformer | 27.074 | 29.772 | 18.835 |
| NIERT | 3.473 | 3.947 | 2.467 |
| NIERT w/ pretraining | **1.897** | **1.971** | **1.246** |

Table 2: Interpolation accuracy of NIERT and the existing approaches over the TFRD-ADlet dataset

| Interpolation approach | Ratio of observed points | | | | |
|---|---|---|---|---|---|
| | 50% | 60% | 70% | 80% | 90% |
| RNN-VAE | 13.418±0.008 | 12.594±0.004 | 11.887±0.005 | 11.133±0.007 | 11.470±0.006 |
| L-ODE-RNN | 8.132±0.020 | 8.140±0.018 | 8.171±0.030 | 8.143±0.025 | 8.402±0.022 |
| L-ODE-ODE | 6.721±0.109 | 6.816±0.045 | 6.798±0.143 | 6.850±0.066 | 7.142±0.066 |
| mTAND-Full | 4.139±0.029 | 4.018±0.048 | 4.157±0.053 | 4.410±0.149 | 4.798±0.036 |
| NIERT | 2.868±0.021 | 2.811±0.032 | 2.656±0.041 | 2.598±0.078 | 2.709±0.157 |
| NIERT w/ pretraining | **2.831±0.021** | **2.771±0.019** | **2.641±0.052** | **2.539±0.085** | **2.596±0.159** |

Table 3: The relationship between interpolation accuracy (measured using MSE, $\times 10^{-3}$) and the ratio of observed points. Here, we use the PhysioNet dataset as representatives

**Accuracy on the NeSymReS dataset:** As shown in Table 1, on the 1D NeSymReS testset, RBF shows the largest interpolation error (MSE: 215.439). MIR, another approach using explicit basis functions, also shows a high interpolation error of 67.281. In contrast, BANP and TFR-transformer, which use neural networks to learn interpolation, show relatively lower errors (MSE: 14.913, 15.556). Compared with these approaches, our NIERT approach achieves the best interpolation accuracy

(MSE: 8.964). Table 1 also demonstrates the advantage of NIERT over the existing approach on the 2D, 3D, and 4D instances.

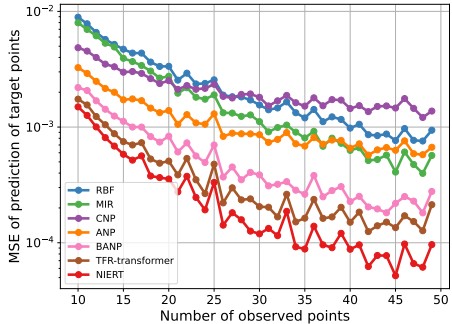

Note that the number of observed points varies greatly in test instances, making the average interpolation error calculated over all test instances insufficient to measure interpolation performance. Therefore, we further divide test instances into subsets according to the number of observed points. As shown in Figure 2, as the number of observed points increases, the interpolation error decreases as expected. In addition, the relative advantages of these approaches vary with the number of observed points, e.g., CNP is better than RBF and MIR initially but finally becomes worse as the number of observed points increases. Among all approaches, NIERT stably shows the best performance over all test subsets, regardless of the number of observed points.

Figure 2: The relationship between the interpolation accuracy and the number of observed points. Here we use the 2D instances in the NeSymReS test dataset as representatives

**Accuracy on the TFRD-ADlet dataset:** As shown in Table 2, CNP, although employing the neural network technique, still performs poorly with MAE as high as of 96.674. In contrast, NIERT achieves the lowest interpolation error (MAE: 3.473), which is over one order of magnitude lower than CNP, ANP, BANP and TFR-transformer. Moreover, when enhanced with the pre-training technique, NIERT can further decrease its interpolation MAE to be 1.897. Besides MAE, other metrics, say CMAE and BMAE, also show the superior of NIERT over the existing approaches (Table 2 ).

**Accuracy on the PhysioNet dataset:** Table 3 suggests that on the PhysioNet dataset, NIERT also outperforms the existing approaches, e.g., when controlling the ratio of observed points to be 50%, NIERT achieves an average MSE ($\times 10^{-3}$) of 2.868, significantly lower than RNN-VAE (13.418), L-ODE-RNN (8.132), L-ODE-ODE (6.721) and mTAND-Full (4.139). Again, NIERT with the pre-training technique shows better performance. The advantages of NIERT hold across various settings of the ratio of the observed points.

Taken together, these results clearly demonstrate the power of NIERT for numerical interpolation in multiple application fields, including interpolating the scattered data generated using mathematical functions, reconstructing temperature fields, and interpolating time-series data.

## 4.3   Case studies of interpolation results

To further understand the advantages of NIERT, we carried out case studies through visualizing the observed points, the reconstructed interpolation functions and the interpolation errors in this subsection. More visualized cases are put in the Supplementary material.

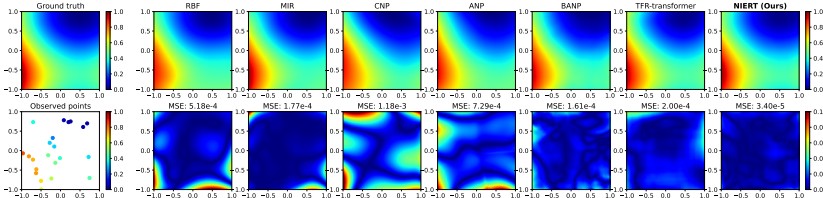

Figure 3: An example of 2D interpolation task extracted from NeSymReS test set. The up-left figure shows the ground-truth function while the bottom-left figure shows the 22 observed points. The interpolation functions reported by NIERT and the existing approaches are listed on the top panel with their differences with the ground-truth are list below

Figure 3 show a 2D instance in the NeSymReS test set, respectively. As illustrated by these two figures, RBF performs poorly in the application scenario with sparse observed data. In addition, RBF and MIR, especially ANP, cannot accurately predict values for the target points that fall out of the range restricted by observed points. The CNP approach can only learn the rough trend stated by the

observed points, thus leading to significant errors. In contrast, BANP, TFR-transformer and NIERT can accurately estimate values for target points within considerably large range, and compared with BANP and TFR-transformer, NIERT can produce more accurate results.

Figure4 shows an instance of temperature field reconstruction extracted from TFRD-ADlet. From this figure, we can observe that when using pre-training technique, NIERT further improves its interpolation accuracy in the whole area.

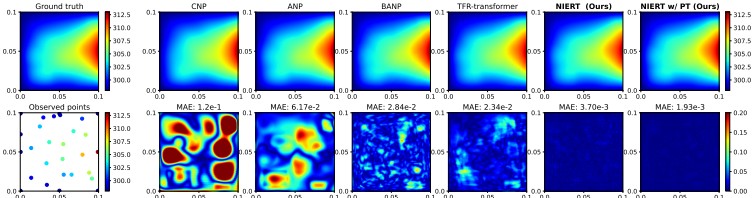

Figure 4: An example of temperature field reconstruction task extracted from TFRD-ADlet test set. The up-left figure shows the ground-truth temperature field while the bottom-left figure shows the 32 observed points. The reconstructed results reported by NIERT and the existing approaches are listed on the top panel with their differences with the ground-truth temperature field are list below

## 4.4 Contribution analysis of observed points for interpolation

An idealized interpolation approach is expected to effectively exploit all observed points with appropriate consideration of relative positions among observed points and target points as well. To examine this issue, we visualized the attention weight of each observed point to all target points. These attention weights provide an intuitive description of the contribution by observed points.

As shown in Figure 5, when using TFR-transformer, the contributions by observed points are considerably imbalanced: on one side, some observed points might affect their neighboring target points in a large region; on the other side, the other observed points have little contributions to interpolation. In contrast, when using NIERT, contributions by an observed point are much more local and thus targeted. More importantly, all observed points have contributions to interpolation.

These results demonstrate that NIERT can exploit the correlation between observed points and target points more effectively.

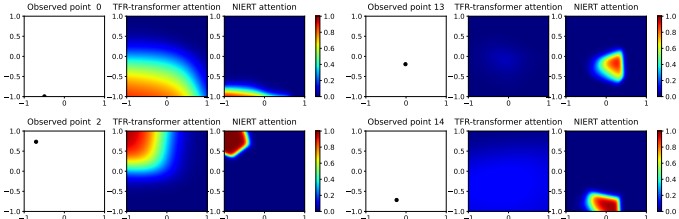

Figure 5: Contributions by observed points for interpolation. The 2D instance is same to that used in Figure 3. We randomly select 4 observed points and extract their attention weights from the final attention layer of NIERT and TFR-transformer. These attention weights provide an intuitive description of the contribution by observed points. The contributions by other 18 observed points are shown in Supplementary material

## 4.5 Ablation study

**The effects of partial self-attention:** For a specific interpolation task, the interpolation function is determined by the observed points only. Thus, an idealized encoding of observed points should not be affected by target points. To investigate the affects of target points, we evaluated NIERT on the test sets with various number of target points. Here, we compared two variants of NIERT, one with partial self-attention, and the other with vanilla self-attention. Both of these two variants were trained using the same training sets (the number of target points varies within [206, 246]).

As illustrated by Figure 6, the variant with vanilla self-attention shows poor performance for the tasks with few target points, say less than 64 target points. In contrast, the variant with partial self-attention always performs stably without significant changes of accuracy.

The results clearly demonstrate that the partial self-attention mechanism allows NIERT to be free from the unexpected affects by the target points.

**The effects of pre-training technique:** To investigate the effects of the pre-training technique, we show in Figure 7 the training process of two versions of NIERT, one without pre-training technique, and the other enhanced with pre-training. As depicted by the figure, even at the first epoch, the pre-trained NIERT shows a sufficiently high interpolation accuracy, which is comparable with the fully-trained BANP and TFR-transformer. Moreover, the performance of the pre-trained NIERT improves in roughly the same convergence speed to the original NIERT. At the final epoch, the pre-trained NIERT decreases the interpolation error to be nearly half of that of the original NIERT.

These results clearly suggest that the experience learned by NIERT from the interpolation task in one application field has potential to be transferred to the interpolation tasks in other application fields.

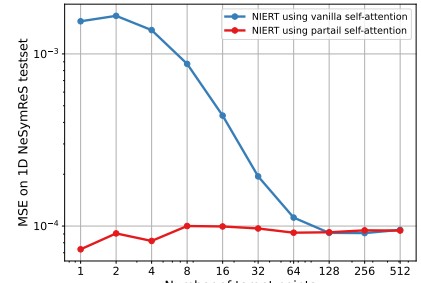 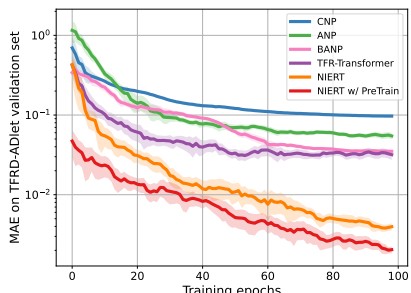

Figure 6: The robustness of NIERT to the number of target points. Here, two variants of NIERT are trained on 1D NeSymReS dataset

Figure 7: The convergence of NIERT, NIERT with pre-training, and the existing approaches. Here, models are trained on TFRD-ADlet dataset

## 5    Discussion and conclusion

We present in the study an accurate approach to numerical interpolation for scattered data. The specific features of our NIERT approach are highlighted by the full exploitation of the correlation between observed points and target points through unifying scattered data representation. At the same time, the use of partial self-attention mechanism can effectively avoid the interference of target points onto the observed points. The enhancement with pre-training technique is another special feature of NIERT. The advantages of NIERT in interpolation accuracy have been clearly demonstrated by experimental results on both synthetic and real-world datasets.

The current version of NIERT has a computational complexity of $O(n(m + n))$, thus cannot handle the interpolation tasks with extremely large amounts of observed points due to the limitations of GPU memory size. Compared with the lightweight traditional methods, our NIERT approach has a much larger model with expensive computation to learn complex function distribution, which limits its application in cost sensitive scenarios. How to reduce the memory requirement and computational cost is one of the future works. Additionally, it is interesting to combine NIERT and the traditional approaches based on basis functions to yield an approach with both high accuracy and interpretability.

We expect NIERT, with extensions and modifications, to greatly facilitate numerical interpolations in a wide range of engineering and science fields.

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
