# NIERT: Accurate Numerical Interpolation through Unifying Scattered Data Representations using Transformer Encoder

## A Broader impact

Numerical interpolation is fundamental in numerical methods and and its has been widely used in a large variety of practical scenarios. In this study, our NIERT approach has been verified in temperature-field reconstruction and time-series data imputation, which shows its potential in weather prediction, industrial environmental monitoring.

## B Theoretical explanation of NIERT

Here we provide a theoretical explanation of NIERT to explain why NIERT performs excellent in tasks of scattered data interpolation. This explanation is based on the tight connection between the core mechanism of NIERT, namely partial self-attention mechanism, and classical RBF interpolation algorithm.

Given $n$ observed points, RBF interpolation formulates the interpolant as

$$\hat{f}(x) = \sum_{j=1}^{n} \lambda_j \phi(x, x_j)$$

where $\phi(x, x_j)$ is the radial basis function related to the observed point $x_j$ and $\lambda_j$ is the coefficient to be determined. In partial self-attention layer of NIERT, a point $x_i$'s representation $\tilde{v}_i$ is computed by

$$\tilde{v}_i = \sum_{j=1}^{n} \alpha(q_i, k_j) v_j$$

where $\alpha(q_i, k_j)$ is the normalized attention weight function. $\alpha(q_i, k_j)$ models the corelation between any query vector $q_i$ ($q_i$ is related to an observed point or a target point) and key vector $k_j$ ($k_j$ is related to an observed point $x_j$).

Obviously, we can regard partial self-attention as a general form of RBF interpolant by corresponding $\alpha(\cdot, \cdot)$ to $\phi(\cdot, \cdot)$ and $v_j$ to $\lambda_j$. Based on neural networks, $\alpha(\cdot, \cdot)$ and $v_j$ is learnable. Thus by adding other appropriate modules and mechanisms, such as MLPs, skip connection, layer normalization, multi-head mechanisms, and applying supervised training, it is promising to get a high-accuracy neural interpolator, which adapts to a certain function distribution.

## C Implementation and experiment details

We provide the details of implementation and experiments in this section and deposit source code of NIERT at `https://anonymous.4open.science/r/NIERT-2BCF`.

Submitted to 36th Conference on Neural Information Processing Systems (NeurIPS 2022). Do not distribute.

**On NeSymReS dataset:**

Table 4 lists the the hyper-parameters of NIERT when running on the NeSymReS dataset. For fair comparison, we set up the TFR-transformer with the same $H$ and $d_{model}$, and the number of its encoder layers is set to 5 and that of its decoder is set to 1 following [1], which shows its best performance. We trained NIERT and the neural approaches for comparison on two NVIDIA GeForce

| Parameter name | Symbol | Value |
|---|---|---|
| Number of layers | $L$ | 6 |
| Hidden dimension | $d_{model}$ | 512 |
| Number of heads | $H$ | 8 |
| $x$'s embedding dimension | $d_{xemb}$ | $16 \times d_x$ |
| $y$'s embedding dimension | $d_{yemb}$ | 16 |

Table 4: Hyper-parameters of NIERT in experiments on NeSymReS dataset

RTX 3090 GPUs for 160 epochs with a batch size of 128. Adam optimizer with a learning rate of $1.0 \times 10^{-4}$ and no schedules are performed for parameters optimization.

**On TFRD-ADlet and PhysioNet dataset:**

For experiments on TFRD-ADlet and PhysioNet dataset, we use the NIERT implementation whose hyper-parameters are listed in Table 5. The pre-trained NIERT also has the same hyper-parameters. Also, for fair comparison, we set up the TFR-transformer with the same $H$ and $d_{model}$, and the number of its encoder layers is set to 2 and that of its decoder is set to 1.

| Parameter name | Symbol | Value |
|---|---|---|
| Number of layers | $L$ | 3 |
| Hidden dimension | $d_{model}$ | 128 |
| Number of heads | $H$ | 4 |
| $x$'s embedding dimension | $d_{xemb}$ | 32 |
| $y$'s embedding dimension | $d_{yemb}$ | 16 |

Table 5: Hyper-parameters of NIERT in experiments on TFRD-ADlet dataset

| Parameter name | Symbol | Value |
|---|---|---|
| Number of layers | $L$ | 3 |
| Hidden dimension | $d_{model}$ | 128 |
| Number of heads | $H$ | 4 |
| $x$'s embedding dimension | $d_{xemb}$ | 16 |
| $y$'s embedding dimension | $d_{yemb}$ | 592 |

Table 6: Hyper-parameters of NIERT in experiments on PhysioNet dataset

For experiments on TFRD-ADlet dataset, we trained NIERT and the approaches for comparison on one NVIDIA GeForce RTX 3090 GPU for 100 epochs with a batch size of 5. Adam optimizer with a learning rate of $5.0 \times 10^{-4}$ with a decay rate of 0.97 is performed. For experiments on PhysioNet dataset, we trained NIERT on one NVIDIA GeForce RTX 3090 GPU for 160 epochs with a batch size of 32. Adam optimizer with a learning rate of $5.0 \times 10^{-4}$ and with a decay rate of 0.97 is performed.

Pre-trained NIERT on 2D NeSymReS is used to fine-tune on TFRD-ADlet dataset. To deal with the problem that the data distribution is too different between NeSymReS data and temperature field in TFRD-ADlet, we simply normalize the data of temperature field to the value range of 2D NeSymReS data before fine-tuning.

To apply NIERT on PhysioNet dataset in which each instance has so many dependent variables with a number of 37, we increase the number of MLPs for dependent variables to 37 in the embedding layer to embed them separately. The pre-training instances for these tasks are correspondingly different. For convenience, we stack 37 random 1D functions in NeSymReS dataset as an instance for pre-training. Although the data generated using such construction method is considerable different from PhysioNet data, we still observe that pre-training technique improves the interpolation accuracy as shown in Table 3.

# D   Dataset details

We evaluated NIERT approach using three datasets, including two synthetic datasets NeSymReS and TFRD-ADlet, and a real-world dataset PhysioNet, which are described in detail as below.

## D.1 Synthetic dataset I: NeSymReS

We used the NeSymReS[1] dataset to test the performance of NIERT for mathematical function interpolation. This dataset was constructed following Biggio et al. [2] with only slight modification for the sake of numerical interpolation. For clarity, we put the details of construction procedure in the next subsection D.2, and list the summarized characteristics of this dataset as follows.

*Instance construction:* We randomly sampled $N$ points from $X = [-1, 1]^{d_x}$ ($d_x$ represents the point dimensionality) and calculate the values for these points with a mathematical function $f$. From these points, we randomly picked $n$ points as observed points and used the left-over as target points. In the study, we chose different $d_x$ for constructing datasets, including 1D, 2D, 3D, and 4D. For 1D NeSymReS dataset, we set the scattered points number $N$ as 256. For 2D/3D/4D NeSymReS dataset, we set $N$ as 512. We control observed points number $n$ within the range of $[5, 50]$.

*Training set and test set:* Each instance in training or test set was built using a mathematical function $f$. These functions were generated using a function sampler as performed in Ref. [2] (see further details in Supplementary text). During the training process, 1 million instances are dynamically sampled at each epoch as training set. We used the $L_2$-form loss function for this dataset. We also generated 12000 instances and used them as test set.

*Dimensionality of points:* We evaluated NEIRT using scatter data with various dimensionality, including 1D, 2D, 3D, and 4D.

*Approaches for comparison:* We compared NIERT with five representative interpolation approaches, including radial basis function (RBF)[3, 4], MIR [5], conditional neural process (CNP)[6], attentive neural process (ANP)[7] and TFR-transformer [1]. Among these approaches, RBF and MIR are classical approaches that use explicit basis functions, while CNP, ANP and TFR-transformer use neural networks to learn how to interpolate.

## D.2 NeSymReS dataset construction details

Following the study of Biggio et al. [2], firstly, we generate equation *skeletons* which is refer to the symbolic equation where numerical constants are replaced by placeholders [2]. Each equation skeleton has the configured number ($d_x$) of independent variables symbols. For example, $y = \sin(C_1 x_1) + C_2 x_2^2$ is a possible generated equation skeleton which includes constants placeholders $C_1$ and $C_2$ and two variables $x_1$ and $x_2$. Such equation skeletons are considered expression trees during generation. Each randomly-generated expression tree has up-to 5 non-leaf nodes, i.e, function operators. Unnormalized weighted distribution shown in Table 7 is used for sampling each non-leaf node. Each leaf node has a probability of 0.8 of being an independent variable and 0.2 of being an integer. Different from [2] using almost all elementary functions symbols including discontinuous ones like $\ln$, $\arcsin$, $\tan$ for symbolic regression tasks, we only use the operators listed in Table 7 which guarantee that the generated function is continuous in the whole domain $X = [-1, 1]^{d_x}$, to make it more suitable for interpolation tasks.

| Operator | $+$ | $\times$ | $-$ | $.^2$ | $.^3$ | exp | sin | cos |
|---|---|---|---|---|---|---|---|---|
| Unnormalized probability | 10 | 10 | 5 | 4 | 2 | 4 | 4 | 4 |

Table 7: Operator and corresponding un-normalized probabilities during the generating process of mathematical functions in NeSymReS dataset

Secondly, constants values are independently sampled from a uniform distribution $\mathcal{U}(1, 5)$ and we get a set of completely-defined mathematical-expression functions. Then we normalize those functions to make their values range in $[0, 1]$ and multiply it with a random number range $[0.9, 1]$ to produce diversity.

After those completely-defined functions obtained, we sample interpolation task for each function by randomly sampling a set of support points $\{x_i\}_i$ in $X$, evaluating the function and get the corresponding $\{y_i\}_i$, and then splitting those data points into a observed points set and a target points

---

[1] Built based on https://github.com/SymposiumOrganization/NeuralSymbolicRegressionThatScales.

97  set. 512 scatter points are sampled using each function, of which a random number (ranging in
98  $[5, 50]$) of points are set up as the observed points and the rest are set up as the target points.

99  During the generation of interpolation tasks, results with invalid values (NaN or inf) are removed.
100 For $d_x$ in configurations $\{1, 2, 3, 4\}$, we generate datasets separately and use them for training and
101 testing. In particular, when $d_x$ is configured, we generate 150 equation skeletons set for testing and 1
102 million skeletons set for training. Each skeleton in the training set existing in the test set has been
103 removed. At training time, interpolation tasks are sampled using equation skeletons from the training
104 set in real time. At testing time, we use an interpolation task set containing 10000 instances, which
105 are generated by the testing skeletons set.

## D.3 Synthetic dataset II: TFRD-ADlet

107 We use the TFRD-ADlet[2] dataset to test the performance of NIERT for 2D temperature field re-
108 construction. This dataset simulates the temperature field of mechanical devices in a high-fidelity
109 fashion[1]. The characteristics of this dataset are summarized as follows. *Instance construction:*
110 Each instance has $200 \times 200$ regular grid points that represent the temperature field in a $0.1m \times 0.1m$
111 square area. Among these grid points, 32 points have their temperate known and used as observed
112 points. The other 3968 points are used as target points.

113 *Training set and test set:* We have a total of 10,000 training instances and 10,000 test instances. For
114 the sake of fair comparison, we also use $L_1$-form loss function for this dataset as performed in Ref.
115 [1].

116 *Dimensionality of points:* The grid points are in a 2D plane.

117 *Approaches for comparison:* For this dataset, we compared NIERT with three neural network-based
118 interpolation approaches, including conditional neural process (CNP)[6], attentive neural process
119 (ANP)[7] and TFR-transformer [1]. We also compared NIERT with its enhanced version that was
120 pre-trained using the NeSymReS dataset.

## D.4 Real-world dataset: PhysioNet

122 We use the PhysioNet[3] dataset, excerpted from the PhysioNet Challenge 2012 [8], to test the
123 performance of NIERT for time-series data interpolation. This real-world dataset were collected from
124 intensive care unit (ICU) records. It should be pointed out that this dataset is a representative of hard
125 interpolation tasks due to the sparsity and irregularity of the records.

126 *Instance construction:* Each instance consists of multiple points, each of which represents a mea-
127 surement of a patient at a specific time. Following the study in Ref. [9], we randomly divided the
128 points into observed points and target points. We also set the ratio of observed points at five levels,
129 i.e., 50%, 60%, 70%, 80%, and 90%, and evaluated NIERT using the thus-acquired datasets.

130 *Training set and test set:* We randomly divided the 8,000 instances acquired from the PhysioNet
131 Challenge 2012 into training set and test set with a ratio of 4:1. For the sake of fair comparison, we
132 use $L_2$-form loss function for this dataset as performed in Ref. [9].

133 *Dinsionality of points:* Each point in an instance represents a measurement at a specific time and each
134 measurement contains up to 37 physiological indices; thus, the independent variable $x$ is 1D while
135 the dependent variable $y$ has a dimensionality of 37.

136 *Approaches for comparison:* On this dataset, we compared NIERT with four representative approaches
137 designed for time-series data interpolation, including RNN-VAE [10], L-ODE-RNN [11], L-ODE-
138 ODE[12], and mTAND-Full[9]. The details of these approaches are provided in Supplementary
139 text. We also compared NIERT with its enhanced version that was pre-trained using the NeSymReS
140 dataset.

---

[2]TFRD-Alet is downloadable at https://pan.baidu.com/s/14BipTer1fkilbRjrQNbKiQ, password: 'tfrd'.
[3]PhysioNet is downloadable at https://physionet.org/content/challenge-2012/1.0.0/.

# E   Approaches for comparison

**Radial basis function (RBF)**   RBF is one of the most commonly-used scattered data interpolation methods. It adopts a specific type of radial basis functions on observed points and uses their linear combination to represent the target function. We use the RBF interpolation implementation in SciPy [13] and multiquadric function as basis function type for the experiments.

**MIR**   MIR [4] is another multivariate interpolation and regression method for scattered data sets proposed by [14]. MIR represents the approximation error with Taylor expansions and minimizes the approximation error to find the basis functions.

**Conditional nerual process (CNP)**   CNP proposed by [6] is an neural model able to learn to predict distributions of target points values given a series of observed points. In order to fully verify the accuracy of interpolation, we let CNP to predict the values of target only and the training criteria function is set to be the prediction error of values of target points in the experiments.

**Attentive nerual process (ANP)**   ANP [7] leverages attention mechanism in CNP and improves the prediction performance. In the experiments, the criteria function are fixed as same as CNP above.

**Bootstrapping attentive nerual process (BANP)**   BANP [15] employs bootstrap technique to further improve the performance of ANP. In the experiments, the criteria function are fixed as same as CNP and ANP above.

**TFR-Transformer**   Transformer [7] is originally proposed to solve tasks in natural language processing. [1] adopts Transformer in 2-dimensional temperature field reconstruction using scattered observations. Compared with vanilla transformer, TFR-Transformer removes positional encoding, encodes the observations using encoder, using cross-attention mechanism between observations' encoding and target points to represents targets' features at decoder, and using a MLP to map targets' features to values.

**RNN-VAE**   RNN-VAE is a VAE-based model where the encoder and decoder are standard RNN models. Gated Recurrent Unit (GRU) [10] module is configured as the recurrent network.

**L-ODE-RNN**   L-ODE-RNN refers to latent neural ODE model where the encoder is an RNN and decoder is a neural ODE proposed in [11].

**L-ODE-ODE**   [12] proposes ODE-RNN model which generalize RNNs to have continuous-time hidden dynamics defined by ODEs. L-ODE-ODE refers to the model where the encoder is an ODE-RNN and decoder is a neural ODE.

**mTAND-Full**   [9] proposes mTAND-Full for interpolation and classification of sparse, irregularly sampled, and multivariate time series data. mTAND-Full performs time attention mechanism to learn temporal similarity and Bidirectional RNNs to encode temporal features. Mask mechanism makes the representation of missing data and target points convenient and easy to parallel.

# F   Additional results

## F.1   Experiments on high-dimensional synthetic data

In order to verify the scalability of NIERT on higher dimensional data, we specially constructed a synthetic 10D dataset to evaluate. In this dataset, each function is obtained by the summation of $K$ randomly sampled 10-dimensional Gaussian functions, which can be formalized as

$$f(x) = \sum_{k=1}^{K} A_k \exp\left(-\frac{1}{2}\frac{(x - c_k)^2}{\sigma_k^2}\right).$$

We fix $K$ as 5. For each Gaussian function using in each function, we uniformity sampled the center $c_k$ from $[-1, 1]^{10}$, width $\sigma_k$ from $[1, 2]$ and weight $A_k$ from $[-1, 1]$.

We created a training set containing 256K instances and test set containing 512 cases from this function distribution. Each instance includes 64 observed points and 192 target points, which are uniformly sampled from $[-1, 1]^{10}$. On the test set, we directly evaluated the classical methods,

---

[4]MIR's implementation can be found at `http://web.mit.edu/qiqi/www/mir/`

and also evaluated the data-driven models after 100 epochs of training. We show the interpolation accuracy of these methods in the Table 8.

| Approach | RBF | MIR | CNP | ANP | BANP | TFR-transformer | NIERT |
|----------|-----|-----|-----|-----|------|-----------------|-------|
| Accuracy | 181.744 | 161.474 | 35.623 | 12.578 | 12.077 | 7.465 | **5.496** |

Table 8: Interpolation accuracy (MSE $\times 10^{-4}$) on 10D test set

These results show that NIERT still maintains the best accuracy on high-dimensional data, suggesting its scalability. Noted that we only changed the input layer dimension of NIERT in order to deal with this 10-dimensional data, so the calculation cost increases by a limited margin.

**F.2   More Ablation studies**

**The effects of different model depths**

We carried out experiments on 2D NeSymReS dataset using NIERT with hyper-parameter $L$ varying from 3 to 7. Then evaluate the models on the 2D NeSymReS test dataset. Accuracy are listed in below Table 9. The results show that NIERT with 7 transformer layers has the best accuracy on the test set, and NIERT with 6 transformer layers has reached a comparable level. Therefore, in the experiments on NeSymReS data set, we use $L = 6$ to balance efficiency and accuracy.

| Interpolation approach | Number of transformer layers $L$ | | | | |
|------------------------|------|------|------|------|------|
| | 3 | 4 | 5 | 6 | 7 |
| NIERT | 66.812 | 60.133 | 52.098 | 45.319 | **44.043** |

Table 9: Interpolation accuracy (MSE $\times 10^{-5}$) varying number of transformer layers $L$ in NIERT on 2D NeSymReS dataset

**The effects of different hidden dimensions**

We also carried out experiments on 2D NeSymReS dataset using NIERT with smaller hidden dimensions $d_{model}$, say from 256, 128 and 64. Then evaluate the models on the 2D NeSymReS test dataset. Accuracy are listed in below Table 10. The results show that when the hidden dimension is within 512, NIERT's interpolation accuracy is higher when the hidden dimension is larger.

In this experiment, we fix other super parameters, say model depth $L$ as 6 and number of heads as 8.

| Interpolation approach | Different hidden dimensions $d_{model}$ | | | |
|------------------------|------|------|------|------|
| | 64 | 128 | 256 | 512 |
| NIERT | 106.193 | 72.107 | 51.153 | **45.319** |

Table 10: Interpolation accuracy (MSE $\times 10^{-5}$) varying hidden dimension $d_{model}$ in NIERT on 2D NeSymReS dataset

**The effects of prediction error of observed points in loss function**

To verify the contribution of re-predicting values of the observed points to the interpolation task, we conducted an experiment that puts the prediction error of observed points in the loss function, i.e. only minimizes the estimation error of target points. The results are shown in Table 11 which demonstrates that only minimizing estimation error of target points make NIERT performing poorly. This indicates that re-predicting the value of observed points helps NIERT to predict the value of target points more accurately.

**The effects of partial self-attention**

| Loss contains prediction error of | Only target points | All points |
|---|---|---|
| MSE ($\times 10^{-5}$) | 48.931 | **45.319** |

Table 11: Interpolation accuracy (MSE $\times 10^{-5}$) of NIERT trained with loss only containing the prediction error of the target points

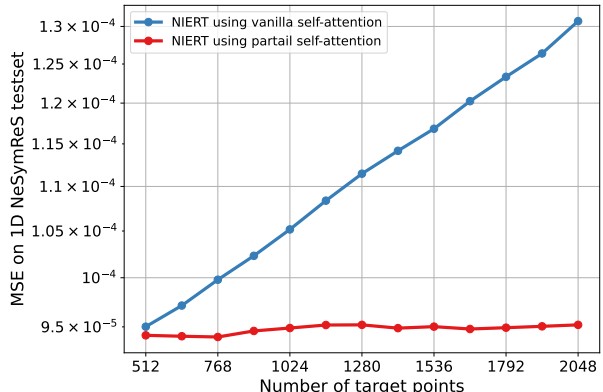

Figure 8: The robustness of NIERT to the number of target points. Here, two variants of NIERT are trained on 1D NeSymReS dataset

To verify the partial self-attention's robustness to number of target points, Figure 7 has demonstrated that when trained with instances whose number of target points varies within [206, 246], the variant NIERT with vanilla self-attention shows poor performance for the tasks with few target points, say less than 64 target points. However, NIERT with partial self-attention performs stably without significant changes of accuracy.

As supplements to Figure 7, we examined the two variants of NIERT by evaluating them on instances whose numbers of target points are much larger than them of training instances, say, varying in [512, 2048]. As demonstrated in Figure 8, as the number of target points increases (from 512 to 2048), the interpolation error of NIERT with vanilla self-attention increases gradually. However, the interpolation accuracy of NIERT with partial self-attention is almost unchanged, and maintains at a better level.

### F.3 Prediction accuracy gap between observed points and target points

| Prediction accuracy on | MSE ($\times 10^{-5}$) on NeSymReS test sets | | | |
|---|---|---|---|---|
| | 1D | 2D | 3D | 4D |
| Observed points | 1.301 | 4.862 | 3.775 | 2.662 |
| Target points | 8.964 | 45.319 | 77.664 | 72.025 |

Table 12: Prediction accuracy gap between observed points and target points on NeSymReS test set

We carried out an extra experiment for prediction accuracy analysis on observed points and target points. Figure 12 shows that on the observed points, the MSE of prediction is relatively smaller when compared with it of target points as expected.

### F.4 Visualization of the embedding layer of NIERT

**Embedding of observed points**

Visualizing the embedding of observed points will help to understand the encoding learned by NIERT. For the two examples shown in the text (Figure 3 4), we extracted the embedding of their

observed points from the NIERT embedding layer, obtained two principal components using PCA dimensionality reduction, and visualized them in the Figure 9 and Figure 10.

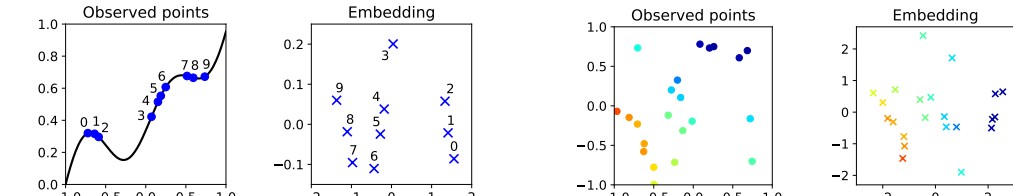

Figure 9: Visualization of scattered data embedding extracted from NIERT's embedding layer using a 1D interpolation case

Figure 10: Visualization of scattered data embedding extracted from NIERT's embedding layer using a 2D interpolation case

From these two examples, we can see that scattered data embedding almost completely retains the relative position information between scattered points. The 1D case also shows that the value information is also embedded as expected (mainly on the second principal component).

**Learned parameters of embedding layer of NIERT**

In addition, we visualized the parameters learned by the niert embedding layer, as shown in the Figure 11 and Figure 12, which correspond to the trained NIERT models on NeSymReS 1D and 2D datasets respectively. These parameters, including weight $w$ and bias $b$ of $\text{Linear}_x$ and $\text{Linear}_y$, and the embedding of masked value $\text{Mask}_y$, indicate how the input scattered data is embedded into the high-dimensional space.

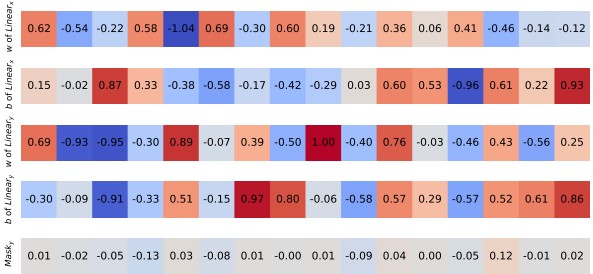

Figure 11: The learned parameters of embedding layer of NIERT for the NeSymReS 1D dataset

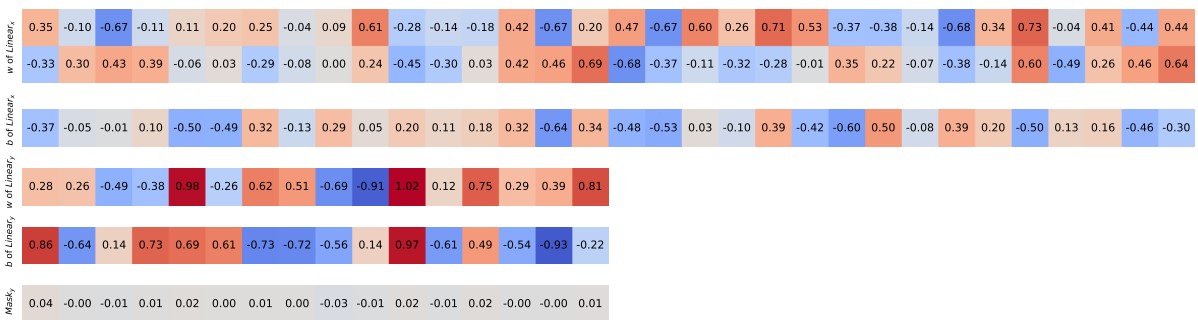

Figure 12: The learned parameters of embedding layer of NIERT for the NeSymReS 2D dataset

 **F.5   More case studies**

 **Cases from 1D & 2D NeSymReS test set**

 In each example of 1D interpolation task extracted from NeSymReS test set, the blue curve represents
 the ground-truth function while the red curves represent the interpolation functions reported by
 NIERT and the existing approaches.

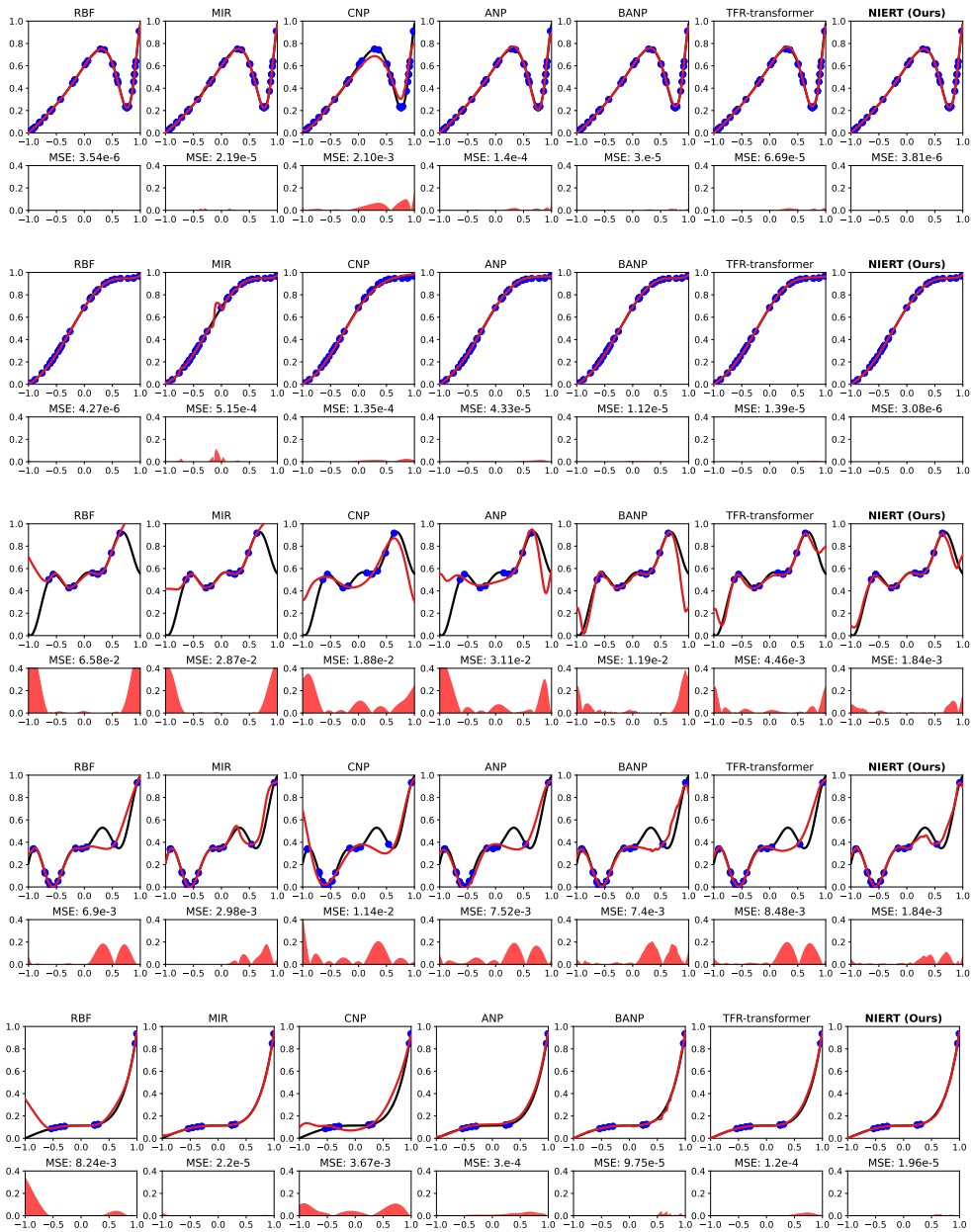

Figure 13: More cases from 1D NeSymReS test set

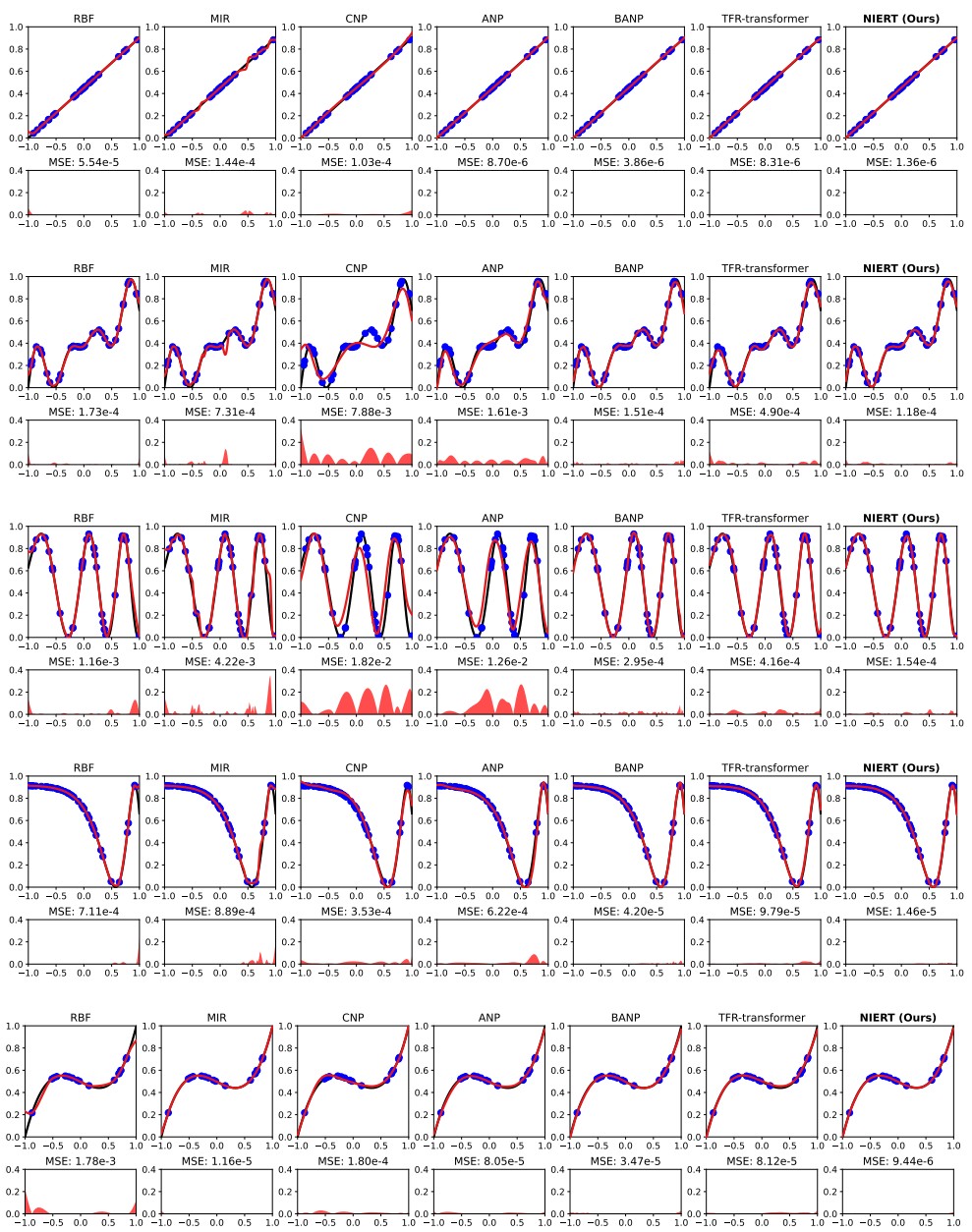

Figure 14: More cases from 1D NeSymReS test set

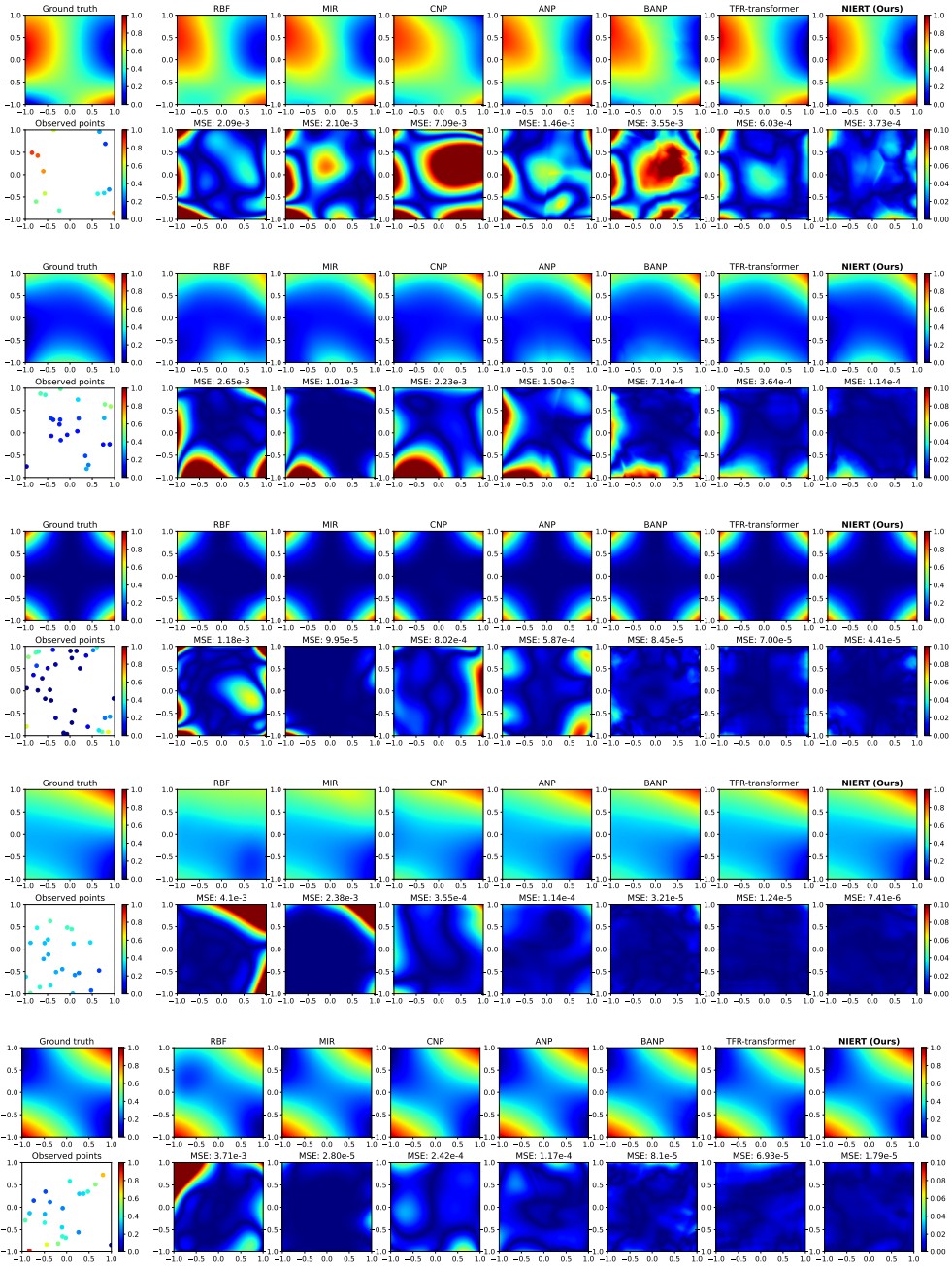

Figure 15: More cases from 2D NeSymReS test set

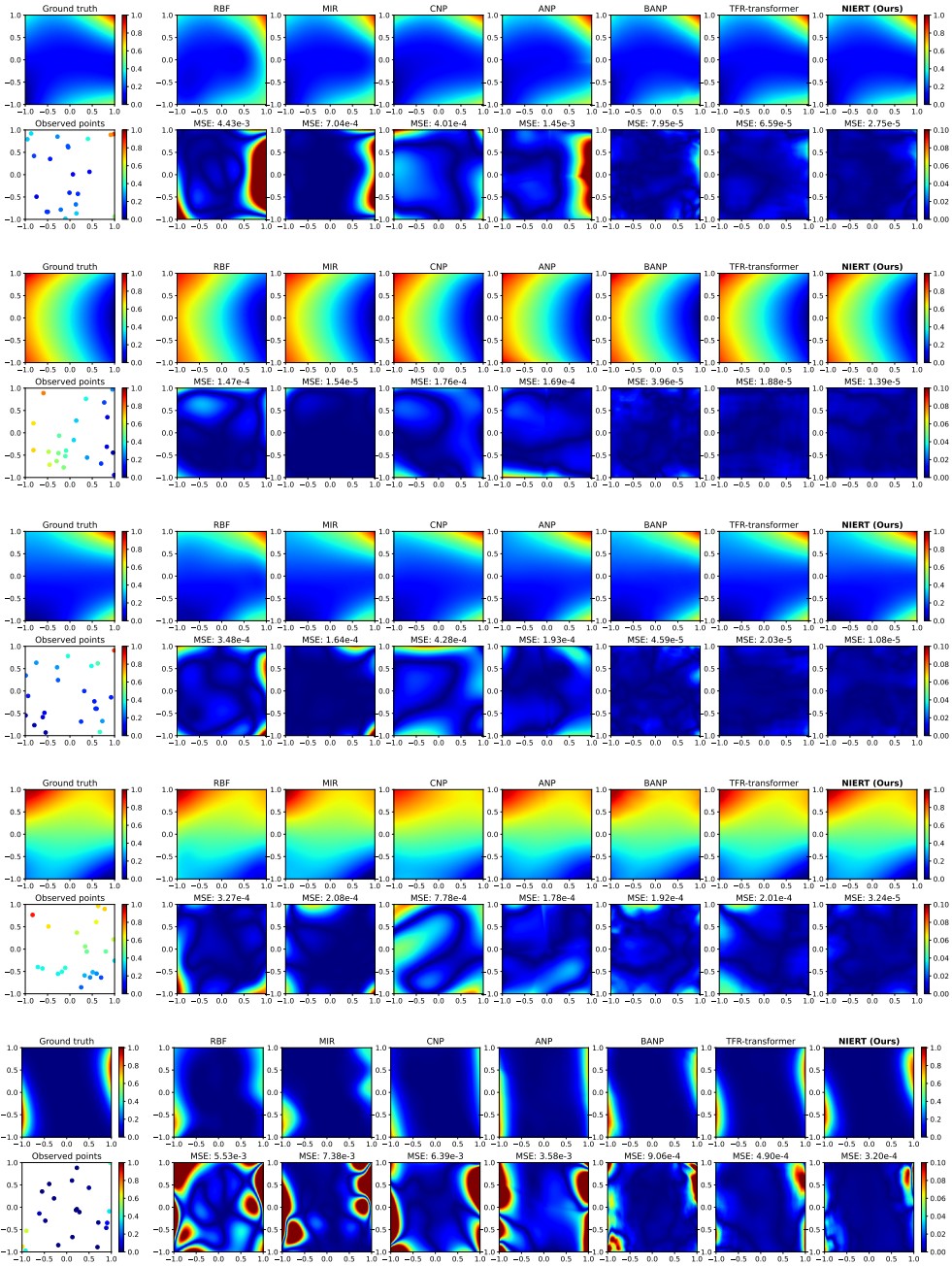

Figure 16: More cases from 2D NeSymReS test set

 **Cases from TFRD-ADlet**

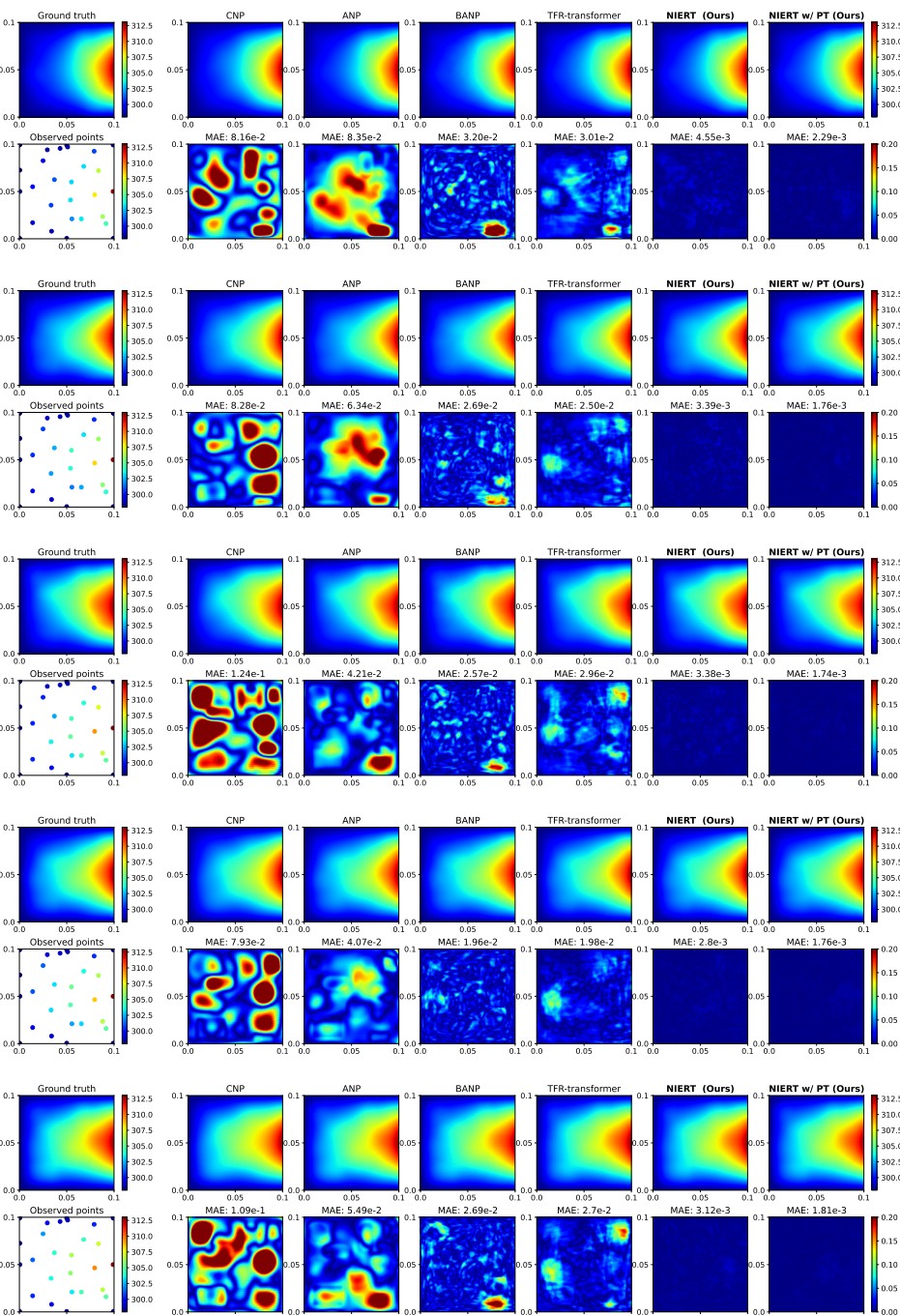

Figure 17: More cases from TFRD-ADlet test set

## F.6 Contribution analysis of observed points for interpolation

As supplements to Figure 6, all observed points' attention weights extracted from the final attention layer of NIERT and TFR-transformer are visualized in Figure 18.

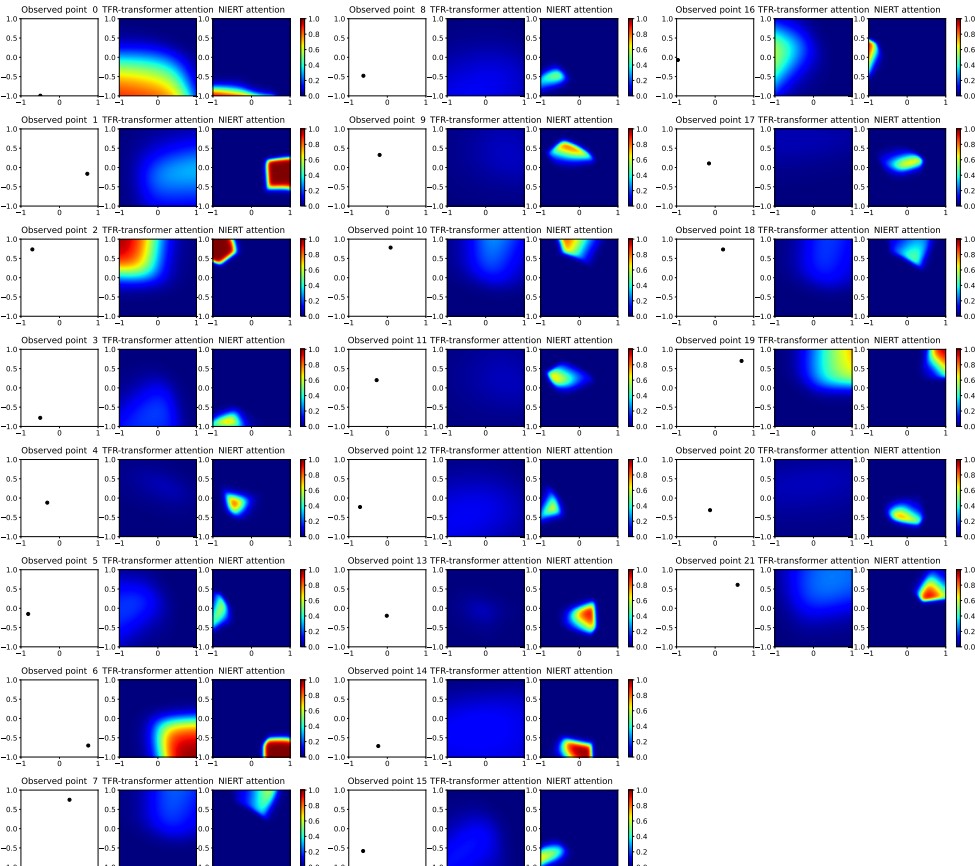

Figure 18: All observed points and the corresponding extracted attention response

The contributions by observed points are quit imbalanced using TFR-transformer. In contrast, when using NIERT, contributions by an observed point are much more local and thus targeted and all observed points have contributions to interpolation. This shows that NIERT can fully exploit the relationship between observed points and target points.