# OpenReview forum: "NIERT: Accurate Numerical Interpolation through Unifying Scattered Data Representations using Transformer Encoder"
_NeurIPS.cc/2022/Conference — NeurIPS 2022 Submitted_

### Official Review · Reviewer_93ZQ · 2022-07-12

**Rating:** 6
**Confidence:** 4
**Soundness:** 3 good
**Presentation:** 3 good
**Contribution:** 3 good

**Summary:**

This is a work on numerical interpolation using transformers. The domain of interpolation is 1d-4d and extends to higher dimensions naturally. An input sequence consists of the following: a function is sampled from a distribution comprising a small set of polynomials, sinusoids and exponentials; input points are sampled uniformly from a cube [-1,1]^d; some of them are deemed observed and the rest are said to be target points (akin to masked inputs). Input points are embedded into a 16-dimensional space using a simple linear layer (Wx + b).


**Questions:**

1. How is RBF tuned in the experiments? In some experiments (say Figure 11), it seems the kernel bandwidth is too large. Also, in general, how are other competing methods tuned?

2. Some errors at the boundaries incurred by RBF could be mitigated by using local polynomial kernels. There are also adaptive basis splines (R mgcv package, for example). Does it make sense to compare NIERT against such methods? It is possible that these methods are inferior to NIERT, but I think for fair comparison we should compare against the best-known classical methods.

3. It is interesting how the inputs are embedded into a 16-dimensional space. What do these parameters look like for 1d and 2d inputs? It would be good to show these in the paper, with perhaps a plot in the 2d case.

4. The NIERT model is large and the input sequence length is limited. Is it possible to reduce the model size? Sorry if I missed, but I did not see any ablation study justifying why you need such a large model. It should be made clear that the interpolation by NIERT is expensive compared to classical methods, say, via flops/wall-clock-time comparison. At least, it should be acknowledged that expensive computation is a known limitation of the model.

5. In Table 3, the MSE increases for some methods as a larger proportion of the data is observed. Why is that?

6. In Section E.2, the MSE on observed points is much less than that on the target points. Why is that?

7. In 1d, does it make sense to sort within the observed data points and also sort within the target data points?


**Limitations:**

No red flags.

**Strengths And Weaknesses:**

Strengths: The empirical results are in favor of NIERT (the proposed method). It beats some interesting/relevant methods.

Weakness: The number of model parameters in NIERT is high and computation is potentially much more expensive compared to classical methods (e.g.: RBF).

---

> ### Author Response · Authors · 2022-08-02
> **Response to Reviewer 93ZQ  Part 1**
>
> We were pleased that Reviewer 3o5r approves overall the idea of our work and reads our work in detail. The reviewer also raised several fundamental and instructive questions which are addressed below.
>
> > **[Q1]** How is RBF tuned in the experiments? In some experiments (say Figure 11), it seems the kernel bandwidth is too large. Also, in general, how are other competing methods tuned?
>
> **[A1]** For both RBF and MIR we just used the default parameter settings to perform the interpolation, because tuning the parameters for each test instance is intractable. Scipy's RBF interpolator and MIR implemented by its author both provide nice ways to set parameters, e.g, the default RBF kernel width is "the average distance between adjacent given points" when interpolating 1D functions.
>
> For the data-driven interpolation methods, including CNP, ANP, BANP and TFR-transformer, we directly evaluated the trained model on the test set.
>
> We additionally readjusted the width of RBF kernel and evaluated the 1D NeSymReS test set (including cases in Figure 11). We observed that on 1D test set and on average, reducing the kernel width will make the RBF perform worse, and slightly increasing the kernel width will make the RBF perform better. We observed that twice the default kernel width get an MSE (x1e-5) of 142.253 (while the default setting got 215.439). Such a result does not affect our conclusion, and we will update all RBF experiments in the future submission.
>
>
> > **[Q2]** Some errors at the boundaries incurred by RBF could be mitigated by using local polynomial kernels. There are also adaptive basis splines (R mgcv package, for example). Does it make sense to compare NIERT against such methods? It is possible that these methods are inferior to NIERT, but I think for fair comparison we should compare against the best-known classical methods.
>
> **[A2]** We agree with the reviewer that B-spline is quite famous and commonly used for interpolation. To our best knowledge, B-spline and its derived methods can be used to interpolate scattered data in one dimension. However, to interpolate functions defined in higher-dimensional space, such methods require that the observed data is on lattice points. Thus, it is difficult to make such comparisons with higher-dimensional scattered data.
>
> Following the reviewer's suggestion, we did an additional experiment on 1D NeSymReS data to examine the performance of cubic-spline and obtained an MSE (x1e-5) of _110.046_ on testset, which suggest that cubic-spline can produce much more accurate results compared with RBF (215.439), but is inferior to MIR (67.281) and data-driven approaches.
>
>
> > **[Q3]** It is interesting how the inputs are embedded into a 16-dimensional space. What do these parameters look like for 1d and 2d inputs? It would be good to show these in the paper, with perhaps a plot in the 2d case.
>
> **[A3]** We realized that the embedding layer we described is not clear enough and may be misleading. We thank the reviewer for pointing out this. In fact, the embedding layer we implemented includes:
>   1. a learnable parameter $\mathrm{MASK}_y$ which represents the missing value,
>   2. and two linear modules $\mathrm{Linear}_x$ and $\mathrm{Linear}_y$ which embed the position $x$ and value $y$ of a scattered point respectively.
>
> We then concatenate embedding of $x$ and $y$ as the point's embedding.
>
> For both the two linear modules, we set their output dimensionality to be 16 times dimensionality of their inputs, to suit the data of different dimensions. $\mathrm{MASK}_y$'s dimensionality the same as the output dimensionality of $\mathrm{Linear}_y$. For clarity, we use the PyTorch code below to describe the parameters of the embedding layer:
> ```python
> self.linear_x = nn.Linear(dim_x, dim_x*16)
> self.linear_y = nn.Linear(dim_y, dim_y*16)
> self.mask_y   = nn.Parameter(torch.zeros(dim_y*16))
> ```
> For example, for 2D scattered data, we embed $x$ from 2D to 32D and $y$ from 1D to 16D. Then we concatenate them and get a point embedding of 48D.
>
> We have clarified this setting in the Supplementary material (Page 2 Table 4-6) now. In addition, we have also revised the description of the embedding layer in the manuscript (Page 4 Line 144-145), where we imprecisely said that the embedding layer is composed of MLPs before.
>
> We additionally visualized the embedding of observed points using the 1d and 2d examples and the results have been placed in the Supplementary (Page 7 Line 224-241). We use PCA for dimensionality reduction. These results show that the points' embedding preserves the relative position information and is also mixed with the value information.

---

> > ### Comment · Reviewer_93ZQ · 2022-08-08
> > **Thanks to the authors.**
> >
> > I appreciate the authors for their responses and for doing more experiments.
> >
> > [A2] Thanks for trying cubic splines. By adaptive basis splines, I meant something like this: https://stat.ethz.ch/R-manual/R-devel/library/mgcv/html/smooth.construct.ad.smooth.spec.html. This approach may give you a better MSE, but still inferior to your proposed method. I believe this will strengthen the paper.
> >
> > [A3] I meant if you could show/plot the parameters of these embeddings layers. Because they are of size 1x16 or 2x16 (plus may be a bias), I thought it would be interesting to see these matrices.
> >
> > [A4] The timings are surprising; probably a matter of efficient implementation. NIERT/RBF FLOPS ratio is ~10^5 but the time ratio is only 16. While I was not expecting a time that is proportional to FLOPS, I did expect a much higher time ratio.
> >
> >
> > [A6] Doesn't this indicate overfitting?
> >
> > Thanks.

---

> > > ### Author Response · Authors · 2022-08-09
> > > **Response to reviewer 93ZQ**
> > >
> > > We thank the reviewer 93ZQ for the instructive feedback and for approving the experiments we have done is in the rebuttal stage. We address the reviewer's concerns below.
> > >
> > > > **[A2]** Thanks for trying cubic splines. By adaptive basis splines, I meant something like this: https://stat.ethz.ch/R-manual/R-devel/library/mgcv/html/smooth.construct.ad.smooth.spec.html. This approach may give you a better MSE, but still inferior to your proposed method. I believe this will strengthen the paper.
> > >
> > > We thank the reviewer for providing more information on adaptive basis splines to enhance our study. We fully agree with the reviewer's suggestion. Due to time constraints, we will add experiments using adaptive basis splines in the future revision.
> > >
> > >
> > > > **[A3]** I meant if you could show/plot the parameters of these embeddings layers. Because they are of size 1x16 or 2x16 (plus may be a bias), I thought it would be interesting to see these matrices.
> > >
> > > We realized that we had misunderstood the reviewer's suggestion. Following the reviewer's suggestion, we additionally visualized the parameters of the embedding layer, including weight $w$ and bias $b$ of $\mathrm{Linear}_x$ and $\mathrm{Linear}_y$, and the embedding of masked value $\mathrm{Mask}_y$. We put the visualized results in the Supplementary material (Page 8 Line 233-238).
> > >
> > > We find from the visualized results that the embedding of the learned embedding of masked value $\mathrm{Mask}_y$ is close to the zero vector, which is the initialization. We also find that it is difficult to directly recognize interpretable patterns for $w$ and $b$ of the linear layers.
> > >
> > >
> > > > **[A4]** The timings are surprising; probably a matter of efficient implementation. NIERT/RBF FLOPS ratio is ~10^5 but the time ratio is only 16. While I was not expecting a time that is proportional to FLOPS, I did expect a much higher time ratio.
> > >
> > > We agree with the reviewer that this time ratio is due to efficient implementation. The RBF interpolator we used is implemented by NumPy, while our NIERT is implemented by PyToch. NIERT's advantages are high parallelism and the use of a highly optimized PyTorch backend. Therefore, NIERT still has very good performance on CPUs.
> > >
> > > Note that we evaluated these two approaches on two Intel Xeon 5218r processors, including a total of 40 cores. NIERT can make full use of multi-core parallelism, but RBF cannot. For fair comparison, we use a process pool with a size of 40 to perform parallel RBF interpolation on the entire test set and then obtain the per-instance running time. When evaluating NIERT, we set the batch size to 10. Both of these settings make the CPU utilization full.
> > >
> > >
> > >
> > > > **[A6]** Doesn't this indicate overfitting?
> > >
> > > This is not overfitting. We fully understand the concerns of reviewers. We realized that we had not explained clearly before. We will try to clarify this matter below.
> > >
> > > In the PhysioNet interpolation experiment, one of our objectives is to verify the ability of NIERT to learn interpolation under different settings of observed data ratio. Therefore, under each specific ratio setting, we divide each instance into observed points and target points according to this ratio, to construct _both the training set and the test set_.
> > >
> > > _Thus, the PhysioNet interpolation datasets are differently distributed under different ratio settings. They are different tasks. Hence, we cannot say that this indicates overfitting, but the dataset of more observed data may lead to worse model learning._
> > >
> > > We re-summarize the reason why dataset of more observed data may lead to worse model learning: in the task with much more observed points and much fewer target points, it may be difficult to learn to predict the target points' value because the supervised signal of the target points is much weaker. Thus, the learned model may perform poorly when testing, even though it sees more observed points when testing.
> > >
> > > Nevertheless, according to our purpose, it is more meaningful to compare the interpolation accuracy of the different models on a certain observed points ratio. Experiments show that the accuracy of NIERT outperforms all these baselines at all settings of observed data ratio.

---

> ### Author Response · Authors · 2022-08-02
> **Response to Reviewer 93ZQ Part 2**
>
> > **[Q4]** The NIERT model is large and the input sequence length is limited. Is it possible to reduce the model size? Sorry if I missed, but I did not see any ablation study justifying why you need such a large model. It should be made clear that the interpolation by NIERT is expensive compared to classical methods, say, via flops/wall-clock-time comparison. At least, it should be acknowledged that expensive computation is a known limitation of the model.
>
> **[A4]** **Ablation study of model size**: We entirely understand the reviewer's concern about the model size and computational cost. We have shown the effect of NIERT's model depth on NeSymRes 2D dataset in the Supplementary material (also see table below), which suggests that the deeper model obtained higher accuracy when the depth is below 7.
>
> |Model depth|Accuracy (MSE x1e-5)|
> |:-:|:-:|
> |4|60.133|
> |5|52.098|
> |6|45.319|
> |7|**44.043**|
>
> We additionally designed a new experiment to examine the effect of hidden dimension on the accuracy of NIERT and present the results in the table below. These results suggest that a larger model is needed if higher accuracy is pursued when the hidden dimension is less than 512.
>
> |Hidden dim|Accuracy (MSE x1e-5)|
> |:-:|:-:|
> |64|106.193|
> |128|72.107|
> |256|51.153|
> |512|**45.319**|
>
> According to the results of current ablation studys, if we want to pursue higher interpolation accuracy, a large model is necessary. This may be explained by that large capacity of large models enables them to learn more complex data distribution. Now we have put these results of the two ablation studys into Supplementary material (Page 6 Line 188-199).
>
> **Computational cost compared with classical methods**: Following the reviewer's suggestion, we compared the per-instance FLOPs/wall-clock-time of NIERT and classical baselines (RBF & MIR) on the NeSymRes 2D test set in the table below.
>
>
> |Method|Accuracy (MSE x1e-5)|Time on CPU (ms)|Time on GPU (ms)|GFLOPs|
> |:-:|:-:|:-:|:-:|:-:|
> |RBF|347.060|**1.29**|-|8.53e-5|
> |MIR|274.601|73.81|-|5.26e-3|
> |NIERT (hidden dim 256)|51.153|7.02|_5.51_|2.53|
> |NIERT|**45.319**|20.29|10.02|9.74|
>
> These results show that NIERT's computational cost is expensive compared with classical RBF method. We can improve the computing speed of NIERT by reducing the model's size and parallelizing, e.g. the NIERT with a hidden dimensionality of 256. We have acknowledged that expensive computation is a known limitation of the model following the reviewer's suggestion (Page 9 Line 316-319).
>
> > **[Q5]** In Table 3, the MSE increases for some methods as a larger proportion of the data is observed. Why is that?
>
> **[A5]** Models evaluated on PhysioNet datasets in Table 3 are all trained to minimize prediction errors of both observed points and target points. As a larger proportion of the observed data, re-prediction error of observed points accounts for a larger proportion in the loss value. This may make the models more inclined to learn to re-predict the value of observed points more, which leads to a MSE increase of target points prediction.
>
> When there are much more observed points than the target points, how to balance the prediction objectives of observed points and target points is a valuable problem. We will consider further exploration in our future work.
>
>
> > **[Q6]** In Section E.2, the MSE on observed points is much less than that on the target points. Why is that?
>
> **[A6]** Because the values of observed points are given, but the values of target points are missing in the input. Thus, for NIERT, learning to re-predict the value of observed points is much easier.
>
> > **[Q7]** In 1d, does it make sense to sort within the observed data points and also sort within the target data points?
>
> **[A7]** In fact, partial self-attention is permutation equivariant for both observed points and target points. Sorting within the observed data points or sorting within the target data points will not affect the results of representation or interpolation at all.
>
> We additionally evaluated NIERT on the NeSymReS 1D test set by randomly permuting the observed points and target points. The NIERT's interpolation results are unaffected as expected.

---

### Official Review · Reviewer_YVNJ · 2022-07-12

**Rating:** 6
**Confidence:** 4
**Soundness:** 3 good
**Presentation:** 2 fair
**Contribution:** 2 fair

**Summary:**

The paper addresses the problem of scatter data interpolation. For that the authors used the encoder representations of Transformers. Since the observed points and the target points are from the same domain, they processed them in a unifying fashion. The paper used partial self-attention to avoid unexpected interference of target points on observed and other target points. The authors also used pre-training technique to improve accuracy. Experimental results were presented on two synthetic and one real datasets.

**Questions:**

- Can you please explain how this approach is "self-supervised" as mentioned in line 10?
- I found the following claim to be too strong compared to provided evidence, since the reported pre-training was done on a synthetic data only. "These results clearly suggest that the experience learned by NIERT from the interpolation task in one 299 application field can be transferred to the interpolation tasks in other application fields."

**Limitations:**

The novelty comes from unifying observed and target data representation and partial self-attention. This is more like the decoder of a transformer. Though it is a good observation, which resulted in promising performance, the novelty of the paper is limited.

**Strengths And Weaknesses:**

Strengths:
- Problem addressed in the paper in important and has practical significance.
- The ablation study was helpful to understand the improvement contributed by each proposed factor.
- The experimental result on real data is promising.

Weaknesses:
- Contribution from pre-training is not very significant.
- "an" -> "a" in line 146, "points and observed points in "an" unifying fashion".
- It will be good to see performance on more real data.

---

> ### Author Response · Authors · 2022-08-02
> **Response to Reviewer YVNJ**
>
> We thank Reviewer YVNJ for pointing out that the problem we study is "_important and has practical significance_", and we were pleased that the reviewer thinks our "_ablation study was helpful_" and "_experimental result on real data is promising_". We also thank the reviewer for pointing out some typos and misleading expressions.
>
> The reviewer also raised some valuable comments which are addressed below.
>
> > Contribution from pre-training is not very significant.
>
> We understand the reviewer's concern about the contribution of pre-training. Although pre-training is a common technique in the fields of NLP and CV, it has not been applied to the data-driven interpolation problem. To our best knowledge, We are the first to use pre-training technique to transfer interpolation experience to improve accuracy in a new interpolation task domain.
>
> Experimental results show that the pre-training can significantly reduce the interpolation MAE of NIERT on TFR-ADlet dataset (from  3.473 to 1.897). For the PhysioNet dataset, which is a highly sparse and irregularly sampled real-world dataset, pre-training can also improve the interpolation accuracy for all settings of different observed data proportions.
>
> > "an" -> "a" in line 146, "points and observed points in "an" unifying fashion".
>
> This has been changed as suggested.
>
> > It will be good to see performance on more real data.
>
> We totally agree with the reviewer's suggestion that it would be better to experiment on more real datasets. We also made efforts to find real-world datasets, and chose PhysioNet because of its scattered nature and did experiments on it in our study. We will continue to look for more suitable real datasets for future evaluation.
>
> > **[Q1]** Can you please explain how this approach is "self-supervised" as mentioned in line 10?
>
> **[A1]** We thank the reviewer for pointing out the potential confusion that is caused by the usage of "self-supervised". Now we have revised them to "learning-based" or "data-driven".
>
> We stated that our work is "self-supervised" before because we considered that NIERT is supervised to learn from scattered data with masked values and to re-predict the values of these scattered data, which is similar to BERT. Besides, we considered that the learned scattered data representation has the potential to be used for some downstream tasks, including function classification, symbolic regression. Since our work focuses on the interpolation tasks, we changed this expression.
>
> > **[Q2]** I found the following claim to be too strong compared to provided evidence, since the reported pre-training was done on a synthetic data only. "These results clearly suggest that the experience learned by NIERT from the interpolation task in one 299 application field can be transferred to the interpolation tasks in other application fields."
>
> **[A2]** We agree with the reviewer and we have revised it into "_These results clearly suggest that the experience learned by NIERT from the interpolation task in one application field has the potential to be transferred to the interpolation tasks in other application fields_" (Page 9 Line 302-303 now).
>
>
> > **[Limitation]**: The novelty comes from unifying observed and target data representation and partial self-attention. This is more like the decoder of a Transformer. Though it is a good observation, which resulted in promising performance, the novelty of the paper is limited.
>
> **[A]** We understand the reviewer's concerns about the novelty and similarity between our NIERT and Transformer decoder. We agree that NIERT model and Transformer decoder are similar, but NIERT still has fundamental differences. We realized that we didn't describe such differences clearly and describe them below now:
> - _Identical embedded space_: Transformer decoder mixes two embedding sequences which belong to different embedded spaces. Differently, in NIERT, the embedding of observed and target points belongs to an _identical_ embedded space. Thus, information passing from observed points to target points will be more effective and accurate.
> - _Self-attention between observed points_: Additionally, partial self-attention of NIERT performs self-attention between observed points at the same time. In this way, the learned correlation between observed points can directly help to model the correlation between observed points and target points, and further improve the quality of the learned representation of target points.
>
> We think the description of the differences between Transformer decoder and NIERT is instructive and we will add these descriptions in the future submission.

---

### Official Review · Reviewer_3o5r · 2022-07-18

**Rating:** 5
**Confidence:** 3
**Soundness:** 2 fair
**Presentation:** 2 fair
**Contribution:** 2 fair

**Summary:**

In this work, the authors propose an accurate approach for numerical interpolation problems. The proposed NIERT method leverages the benefits of the self-supervised mechanism and pre-training technique to improve accuracy.  Several benchmark datasets are employed here to illustrate the effectiveness, compared with the baseline methods. Also, the authors provide some discussions on limitations and future plans, specifically, computational cost and interoperability.

**Questions:**

1. numerical integration is important and several mathematical ways are good enough, what's the real motivation? why do you pursue a higher accuracy, is that necessary enough?  what's the real impact on a specific application?  I just saw a very general description but did not very understand the motivation.

2. Most of the baselines are not competitive. I think the only strong one is TFR[2], what are the benefits compared with this method? why NIERT is better than that if both use transformer-based structures?

3. Just wonder, this work is like transformer + X, kind of work, but how do you think the transformer will work better?  in other words, the study lack interpretability. Could you highlight the origin of the idea?

4. About the correlation, I did not get how to explore the correlation, could you give more details?

5. The algorithm flowchart is not informative, could you polish it a little bit to show more details?

6. The benchmarks mainly focus on 1d and 2d examples, how about the scalability of this algorithm?  I think the real challenge is high-dimensional numerical integration. If the algorithm can handle that, what's the accuracy-cost trade-off?



**Ethics Review Area:**

["I don’t know"]

**Limitations:**

yes

**Strengths And Weaknesses:**

Strength

1. interesting topic and well-structured contexts
2. solid benchmark and baseline comparisons

Weaknesses

1. Lack of real-world high-dimensional applications
2. Motivation is not clear, only for numerical integration? any benefits for large-scale AI applications?
3. Something like transformer + X work, what's the core contribution and benefits?

---

> ### Author Response · Authors · 2022-08-02
> **Response to Reviewer 3o5r Part 1**
>
> We were pleased that Reviewer 3o5r approves overall the idea of our work and thinks our study has an "_interesting topic_" and "_solid benchmark and baseline comparisons_". The reviewer also raised several fundamental questions and constructive comments which are addressed below.
>
> > **[Q1]** numerical integration is important and several mathematical ways are good enough, what's the real motivation? why do you pursue a higher accuracy, is that necessary enough? what's the real impact on a specific application? I just saw a very general description but did not very understand the motivation.
>
> **[A1]** We realized that we did not clearly describe the motivation of our study. The real motivation of our study comes from the observation that, despite the success of the classical mathematical methods of scattered-data interpolation for general applications, they still suffer from several limitations, including the high requirement of sufficient observed points, and the limited complexity of the target function.
>
> The continuous improvement in the accuracy of numerical scattered-data interpoation will yield a significant impact in practical application field. For example, temperature field reconstruction for micro-scale electronics, as a base task to obtain the real-time working environment of electronic components, is one of the effective approaches of health detection system. Accurate data-driven interpolation methods will help determine the working status of the devices from observations of a limited number of sensors, then adjust the operative mode timely, and further considerably improve the durability and reliability of the electronic devices and reduce the maintenance costs in the systems [1].
>
> We have now inserted sentences into the manuscript to describe the real motivation (Page 1 Line 29-32, Page 2 Line 38-39).
>
> > **[Q2]** Most of the baselines are not competitive. I think the only strong one is TFR[2], what are the benefits compared with this method? why NIERT is better than that if both use Transformer-based structures?
>
> **[A2]** We completely understand the concerns of the reviewer on the selection of baseline approaches. In the study, we chose these comparable classical methods and data-driven methods for scattered-data interpolation to the best of our ability. Among them, RBF is one of the most popular classical methods. MIR is a high-precision non-learning-based interpolation algorithm, which is the most recently proposed and open source. CNP, ANP, BANP, TFR-transformer are a class of new methods based on neural networks, which can learn interpolation from certain datasets.
>
> We also completely agree with the reviewer in that TFR-transformer is the strongest baseline approach. The differences between our NIERT approach and TFR-transformer include:
>
>    1. _Architecture_ : TFR-transformer consists of a Transformer encoder and a Transformer decoder and treats the observed points and target points separately, i.e, the encoder encodes the observed data and the decoder produces target points' representation through cross-attention with observed points' encoding. Differently, NIERT adopts only a Transformer encoder (equipped with partial self-attention) to encode and learn correlation of the scattered data in a unified fashion. In this way, target points and observed points are processed identically in each layer and their representations can be learned through message passing in the same feature space. Thus, the information in the learned representation of target points will be more sufficient and accurate.
>
>   2. _Training objective_ : TFR-transformer is trained by minimizing the prediction error of only target points. Benefiting from a unified structure, we trained NIERT by minimizing the prediction error of both observed points and target points. Re-predicting the value of observed points is much easier than interpolating the values of target points. Training with this objective can help NIERT to learn the representation of observed points and their correlation. Such representation and relationship is transferred to the target points through message passing in the identical feature space, so as to improve the prediction accuracy of the target points.
>
> We think the description of the difference between TFR-transformer and NIERT is instructive and we are now pleased to expand the description to illustrate this point (Page 3, Line 95-100).
>
>
> [1] Chen, Xiaoqian, et al. "Tfrd: A benchmark dataset for research on temperature field reconstruction of heat-source systems." arXiv e-prints (2021): arXiv-2108.

---

> ### Author Response · Authors · 2022-08-02
> **Response to Reviewer 3o5r Part 2**
>
> > **[Q3]** Just wonder, this work is like Transformer + X, kind of work, but how do you think the Transformer will work better? in other words, the study lack interpretability. Could you highlight the origin of the idea?
>
> **[A3]** We agree with the reviewer that our work NIERT is a bit like "_Transformer + X_". However, it should be noted that NIERT has core differences from vanilla Transformer:
>   1. The main component of NIERT only has an encoder.
>   2. We modified self-attention to partial self-attention in this Transformer encoder to meet the inductive bias of interpolation.
>
> We interpret our NIERT via a tight connection with the classical RBF approach. Let's have a look first at the formalizations of these two approaches which are listed below:
>
>   - _RBF_: RBF interpolation formulates the interpolant as
>     $$f(x)= \sum_j\lambda_j\phi(x,x_j)\tag{1}$$
>     where $\phi(x,x_j)$ is the radial basis function related to the observed point $x_j$ and $\lambda_j$ is the coefficient.
>
>   - _NIERT_: In the core mechanism of NIERT, namely partial self-attention layer, a point $x_i$'s representation $\tilde{v}_i$ is computed by
>     $$\tilde{v}_i = \sum_j\alpha(q_i,k_j)v_j\tag{2}$$
>     where $\alpha(q_i,k_j)$ is the normalized attention weight function. $\alpha(q_i,k_j)$ models the corelation between any query vector $q_i$ and key vector $k_j$ ($k_j$ is related to an observed point $x_j$).
>
>   We can easily find that Eq.(2) is a general form of Eq.(1) by corresponding $\alpha(\cdot,\cdot)$ to $\phi(\cdot,\cdot)$ and $v_j$ to $\lambda_j$. Thus, by enhancing with other mechanisms, such as layer normalization, skip connection and multi-head mechanism, and applying supervised training, it is promising to obtained a high-accuracy and generalizable neural interpolator.
>
> The above deep connection is one of the origins of our idea. Another origin of our idea is that masked language model like BERT, which can predict missing tokens based on some given tokens in a sentence, which resembles interpolation problem.
>
> We think the tight connection between our NIERT and RBF interpolation is also instructive. Due to space limitations, we expand these explanations in the Supplementary material (Page 1, Line 6-18).
>
> > **[Q4]** About the correlation, I did not get how to explore the correlation, could you give more details?
>
> **[A4]** We define _correlation_ of an observed point and a target point as _the learned attention score between them_ in NIERT model, namely $\alpha(q_i,k_j)$ decribed in [A3], since it represents the proportion of information passing from an observed point to a certain target point.
>
> We showed such correlation of each observed point on the entire domain, namely $\alpha(\cdot,k_j)$ (Figure 5 of revised paper & Figure 16 of revised Supplementary material). For each observed point, we extract the head with the highest response from the last multi-head partial self-attention layer of NIERT.
>
> These results show that the correlation is very similar to Gaussian RBF, that is, each observed point only affects the area near it. Meanwhile, unlike Gaussian, it is non-centrosymmetric and adaptive. These implies the interpretability of NIERT decribed in [A3].
>
> > **[Q5]** The algorithm flowchart is not informative, could you polish it a little bit to show more details?
>
> **[A5]** We agree with the reviewer that our flowchart needs to be polished. We will update the polished flowchart in the future submission.
>
> > **[Q6]** The benchmarks mainly focus on 1d and 2d examples, how about the scalability of this algorithm? I think the real challenge is high-dimensional numerical integration. If the algorithm can handle that, what's the accuracy-cost trade-off?
>
> **[A6]** We completely agree that high-dimensional interpolation is much more challenging and understand the reviewer's concern about the scalability of our approach NIERT. We also tried to evaluate the NIERT in higher dimensional dataset. The lack of real-world high dimensional data leads us to synthetic data. However, due to the complexity of synthetic high-dimensional symbolic functions, we only evaluated 3D and 4D NeSymReS dataset in our study.
>
> Following the reviewer's suggestion, we additionally build a new 10-dimensional function set, where each function is obtained by the summation of several randomly sampled 10-dimensional Gaussian functions. We randomly set the center, width, and weight of each Gaussian function. The details have been added to the Supplementary material (Page 5 Line 175-186) and the results listed in table below show that NIERT still obtains the highest interpolation accuracy in high-dimensional datasets, which shows its scalability.
>
> |Method|Accuracy (MSE x1e-4)|
> |:-:|:-:|
> |RBF|181.744|
> |MIR|161.474|
> |CNP|35.623|
> |ANP|12.578|
> |BANP|12.077|
> |TFR-transformer|7.465|
> |NIERT|**5.496**|
>
> In this experiment, we only changed the dimension of the input layer to adapt to 10-dimensional scattered data. Hence, the calculation cost increases very slightly.

---

### Meta-Review · Area_Chair_YySZ · 2022-08-26

**Recommendation:** Reject
**Confidence:** Less certain

**Metareview:**

This work proposed an accurate approach that called NIERT for numerical interpolation problems. The motivation of the proposed NIERT is not explained very well. The major limitation lacks of  real-world  applications in high-dimensional datasets. Due to the main concern of the work, this shoud be necessary.

**Award:**

No

---

### Decision · Program_Chairs · 2022-09-14

Reject